# Reported Outcome Measures in Studies of Real-World Ambulation in People with a Lower Limb Amputation: A Scoping Review

**DOI:** 10.3390/s22062243

**Published:** 2022-03-14

**Authors:** Mirjam Mellema, Terje Gjøvaag

**Affiliations:** 1Department of Mechanical, Electronic and Chemical Engineering, Faculty of Technology, Art and Design, Oslo Metropolitan University, P.O. Box 4, St. Olavs Plass, 0130 Oslo, Norway; 2Department of Occupational Therapy, Prosthetics and Orthotics, Faculty of Health Sciences, Oslo Metropolitan University, P.O. Box 4, St. Olavs Plass, 0130 Oslo, Norway; terje@oslomet.no

**Keywords:** wearable technology, accelerometer, activity monitor, walking activity, free-living environment, ecological validity

## Abstract

Background: The rapidly increasing use of wearable technology to monitor free-living ambulatory behavior demands to address to what extent the chosen outcome measures are representative for real-world situations. This scoping review aims to provide an overview of the purpose of use of wearable activity monitors in people with a Lower Limb Amputation (LLA) in the real world, to identify the reported outcome measures, and to evaluate to what extent the reported outcome measures capture essential information from real-world ambulation of people with LLA. Methods: The literature search included a search in three databases (MEDLINE, CINAHL, and EMBASE) for articles published between January 1999 and January 2022, and a hand-search. Results and conclusions: 98 articles met the inclusion criteria. According to the included studies’ main objective, the articles were classified into observational (*n* = 46), interventional (*n* = 34), algorithm/method development (*n* = 12), and validity/feasibility studies (*n* = 6). Reported outcome measures were grouped into eight categories: step count (reported in 73% of the articles), intensity of activity/fitness (31%), type of activity/body posture (27%), commercial scores (15%), prosthetic use and fit (11%), gait quality (7%), GPS (5%), and accuracy (4%). We argue that researchers should be more careful with choosing reliable outcome measures, in particular, regarding the frequently used category step count. However, the contemporary technology is limited in providing a comprehensive picture of real-world ambulation. The novel knowledge from this review should encourage researchers and developers to engage in debating and defining the framework of ecological validity in rehabilitation sciences, and how this framework can be utilized in the development of wearable technologies and future studies of real-world ambulation in people with LLA.

## 1. Introduction

The use of wearable technology to monitor real-world ambulatory activity in people with a Lower Limb Amputation (LLA) has grown rapidly in the past decade. Activity monitors have the potential to provide objective information about peoples’ ambulatory behavior and participation in the community, an important domain of the International Classification of Functioning, Disability and Health [1]. Data from the Swedish Amputation and Prosthetic Registry show that 27–48% of the 5762 persons with LLA report to not walk outdoors one year post-amputation [2]. Community walking is essential to participate in work, leisure, social activities, and family roles, and the inability to ambulate outside of the home increases isolation and dependency [3]. Monitoring ambulatory behavior of people with LLA in a free-living setting gives valuable information that can be used to develop rehabilitation programs and prosthetic component prescription that are adjusted to the individual’s need.

Traditionally, the prosthetic user themselves informs the clinician about their ambulatory behavior and physical activity level. However, studies report that self-reported prosthetic use and activity level are both overestimated as well as underestimated by the user, indicating that a subjective assessment is unreliable [4]. Alternatively, standardized performance-based tests can be used to objectively assess a person’s physical functioning and to monitor changes over time. However, comprehensive testing is required to obtain a comprehensive picture of a person’s functioning, since single tests such as the 6 Meter Walk Test or the Timed Up and Go test measure different characteristics of functional capacity. Performing multiple tests can be an exhaustive and time-consuming process, yet the outcomes reflect only a snapshot of an individual’s functioning and can be highly influenced by the person’s conditions at that particular moment. Moreover, a clinical test environment can be considered as revealing optimal walking capability under idealistic conditions, which does not necessarily reflect real-world situations [5].

Prior studies in people with LLA found that in-laboratory activities look different from real-world behavior [6]. The real world is the context where ambulation takes place in a free-living, unsupervised, uncontrolled, and non-standardized manner [7]. The individual’s ambulatory behavior depends on the personal situation and contextual and environmental factors, such as terrain and seasonal conditions [8]. Consequently, experiments that do not contain typical characteristics of real-world ambulation are limited in the extent that findings can be generalized to real-life situations. Wearable technologies enable new opportunities for monitoring ambulatory patterns and temporospatial gait characteristics in a real-life setting. Understanding the relationship between in-clinic performance-based measures and real-world ambulatory performance can improve prosthetic recommendation, physical activity goal-setting and follow-up assessment. Hence, wearable technologies are a step forward in bridging the gap between in-laboratory and real-world measurements.

A preliminary search for existing literature reviews on the use of wearable technology in real-world ambulation was conducted in April 2021 in the following databases: Johanna Briggs Institute (JBI) Evidence Synthesis, Cochrane Database of Systematic Reviews, PubMed, MEDLINE, and CINAHL. Identified reviews have investigated the validity and reliability of wearable technologies [9,10], reported the efficacy of wearable activity monitoring on weight loss [11], and physical activity participation [12]. Other reviews evaluated the use of wearable technology to monitor physical activity in patients with Chronic Obstructive Pulmonary Disease (COPD) [13], stroke [14], and community-dwelling adults [15]. A scoping review of Wong et al. [16] mapped the evidence on physical activity level using wearable technology in people with LLA. Chadwell et al. [17] investigated the methodologies and technologies used to assess the use of upper- and lower-limb prostheses and discussed the barriers for use of wearable technology in low-resource settings. However, the latter review did not discuss the different outcome measures that are reported in studies using wearable technologies, and to what extent the outcome measures are representative for what they intend to measure. Hence, there is a knowledge gap that needs to be investigated.

Therefore, this article reviews the scientific literature on the application of wearable technologies to report on real-world ambulation and prosthetic use in people with LLA. For this purpose, we chose to use the scoping review methodology to map, structure, and analyze evidence in this area of research. The objectives of this review are (1) to investigate for what purposes wearable activity monitors are used in people with LLA, (2) to identify and structure the reported outcome measures, and (3) to evaluate to what extent the reported outcome measures capture essential information of real-world ambulation of people with LLA. To the best of our knowledge, this has not been conducted before, and this review will provide new knowledge to the rehabilitation sciences, that can be used as a guide for researchers and practitioners in research of real-world ambulation and the choice of relevant outcome measures for this population.

## 2. Materials and Methods

We utilized the JBI Manual for Evidence Synthesis provided by The Joanna Briggs Institute for developing the scoping review [18], and structured the content according the Preferred Reporting Items for Systematic Reviews and Meta-Analysis—Scoping Reviews (PRISMA-ScR) updated version 2021 [19]. An a priori protocol including the objectives, inclusion criteria and methods for this scoping review was registered on 18 May 2021 in the Open Science Framework (OSF) with registration number DOI 10.17605/OSF.IO/7PF2U. An updated version of the protocol was registered on 23 September 2021 [20].

### 2.1. Search Strategy

The search was conducted in the databases MEDLINE, EMBASE, and CINAHL on 19 May 2021. The initial search including MeSH terms and keywords was developed in MEDLINE through consultation with a certified research librarian at our university, and then modified for application in the EMBASE and CINAHL databases. The search was limited to include records from 1999 until the day of the search. The starting year was based on the first publication that provided a detailed description of a Step Activity Monitor (SAM), used for long-term, continuous recording of ambulatory activity in both normal and impaired gait [21]. We also reviewed the web pages of the wearable technology brands and hand-searched reference lists of included studies to identify additional relevant articles that may have been missed in the database searches. An update search was performed on 7 January 2022. The PRISMA flowchart (Figure 1) was updated with backward correction, according to the step-by-step description by Bramer and Brain [22]. The MEDLINE search is included in Appendix A.

### 2.2. Source of Evidence Screening and Selection

The reference manager software tool EndNote X9 (Clarivate, Philadelphia, PA, USA) was used to export records and to identify and remove duplicates. Remaining records were exported to the web-based analysis tool Rayyan (Rayyan Systems Inc., Cambridge, MA, USA), that allows for blinded screening by multiple reviewers [23]. The screening process involved two phases: (1) reviewing based on title and abstract and (2) full-text article reviewing. For the first phase, two reviewers independently assessed the inclusion eligibility and rated each record with either ‘include’, ‘exclude’, or ‘maybe’. Articles were included if the studies involved (1) at least one person with LLA using a prosthesis, (2) quantitative measurements using wearable technology in a real-world setting, e.g., outside the clinic or laboratory, and (3) participants who were free to decide their ambulatory behavior, without receiving supervision or other instructions. Studies including only persons with partial foot amputation were excluded. Additionally, studies in real-world setting where participants performed a testing protocol or received supervision or other instructions on how to behave, were excluded. Following phase 1, the two reviewers conducted debriefing meetings to discuss potential disagreements on the screening process, that were then solved by consensus. The second phase was performed by one reviewer who reviewed the full text of included articles from phase 1 for inclusion criteria. All included and excluded articles from phase 2 were subsequently discussed with the second reviewer to verify if the two reviewers agreed upon the decision of the first reviewer. Only peer-reviewed articles in the English language were included. Yet, since this is a scoping review, conference abstracts, clinical letters, and Ph.D. theses were included if they met the inclusion criteria. With duplicates existing of an original research article and a conference abstract, the original research article was prioritized, and the conference abstract excluded.

### 2.3. Data Extraction

The following data was extracted from each article: author(s), year of publication, country of publication, title, study design, objective(s), study population (number of participants, age, etiology, and level of amputation), control group if applicable, wearable technology used, placement of technology, environment and duration of real-world measurements, intervention if applicable, outcome measures reported for the used wearable technology, key findings, conclusion, and clinical relevance. The first reviewer initially extracted the data, and the second reviewer verified the data.

### 2.4. Analysis and Presentation of Results

The main objective of each study was used to synthesize categories of study design. The study design categories were used as the main structure of an overview table of all included studies. The following information from the extracted data was listed in Table 1: First author (year and country of publication), title, objective, study population (number of participants (number of females)), age, level and etiology of amputation, technology used (placement on body and duration of monitoring), reported outcome measures, and key findings. The study design categories were described with a detailed summary of the main objective from all articles. The publication year of included articles was summarized and presented in Figure 2. The analysis of reported outcome measures was performed by collecting all reported outcome measures related to the used wearable technology from each included article. The categorization of reported outcome measures was an iterative process of searching for patterns in the large number of outcome measures and discussions between the first and second reviewer.

## 3. Results

### 3.1. Search Results

The initial literature search resulted in 4006 records after removing duplicates. Then, 3198 records were excluded after screening the title and abstract, leaving 115 articles for full-text eligibility screening. Of these, 93 articles met the inclusion criteria, and in addition, 5 articles were identified through hand-searching, resulting in 98 articles included in this review. See details of the literature search and reasons for exclusion in the PRISMA flowchart in Figure 1.

### 3.2. Inclusion of Sources of Evidence

Most studies were performed in the US (*n* = 65), followed by United Kingdom (*n* = 7), The Netherlands (*n* = 6), Australia (*n* = 6), Canada (*n* = 4), Germany (*n* = 3), Norway (*n* = 2), Thailand (*n* = 2), Hong Kong (*n* = 1), Chile (*n* = 1), and Taiwan (*n* = 1). Almost half of the studies were published in the past four years (see Figure 2).

The most frequently used wearable technologies were the StepWatch (Modus Health, Edmonds, Santa Monica, CA, USA) [4,21,25,27,28,30,42,44,46,48,49,50,51,54,57,58,61,64,65,69,73,77,78,79,84,86,87,88,89,90,94,95,96,98,100,101,110,114,115,116], ActiGraph (ActiGraph, Pensacola, FL, USA) [29,31,32,39,41,43,45,47,53,70,75,80,81,82,83,92,105,109], FitBit (FitBit, San Francisco, CA, USA) [34,40,52,76,108,117], and activPAL (PAL Technologies, Glasgow, Scotland, UK) [26,36,37,68,91,103,104]. Nine studies used a custom sensor design [41,60,74,102,106,107,109,112,113], six studies used pedometers [56,59,71,72,99,113], and one monitored physical activity with a mobile phone [118]. Six studies added a Global Positioning System (GPS) device in addition to the activity monitors [54,103,105,110,114]. Other wearable technologies used in the included studies were TEMEC activity monitor (Temec Technologies BV, Heerlen, The Netherlands) [63,67], Dynaport ADL (McRoberts BV, The Hague, The Netherlands) [62,119], Power Walker (Yamax Health & Sports Inc., Shropshire, UK) [85], EmpowerGO (Hanger Inc., Austin, TX, USA) [24], Up move (Jawbone, San Francisco, CA, USA) [35], Activ 8 (VitaMove, Veldhoven, The Netherlands) [38], Uptimer (National Ageing Research Institute, Victoria, Australia) [66], AMP 331 (Dynastream Innovations, Cochrane, AB, Canada) [97], and the MiniMods Dynaport (McRoberts BV, The Hague, The Netherlands) [33]. Two studies used an accelerometer, but did not specify the manufacturer [55,93]. 

The articles were structured into four categories, based on the primary reason for using wearable technology: observational studies (*n* = 46), interventional studies (*n* = 34), algorithm/method development studies (*n* = 12), and validity/feasibility studies (*n* = 6). Of the 46 included observational studies, 13 studies aimed to describe characteristics of real-world behavior [21,26,33,40,60,63,66], real-world measurements in a specific population [44,50,62], or at a specific moment in time, such as post-rehabilitation [21,45,50], or during a mountaineering expedition [59]. Four studies observed changes in real-world behavior over time, including changes between before and during the COVID-19 pandemic [24,27], during in-patient and leave periods [36], and before and after prosthetic disuse [25]. One study investigated the effect of various prosthetic accommodation durations [35]. Additionally, 28 studies investigated the correlation between real-world measurements and other variables, such as performance-based measures [28,33,34,37,38,47,50,53,56,61], self-reported outcomes [4,26,33,35,42,49,61,67], K-level [45,48,58], measures of gait quality [64], fall and injury incidence [30], aerobic capacity [28], demographic factors [26,61], limb fluid volume [43], clinical scores [62], comorbidities [47], stride-to-stride fluctuations [53], step variability [56], and prosthetists’ perceptions of participants’ prosthetic use [41]. Eight studies performed comparisons between groups, based on the etiology of amputation [26], level of amputation [51], K-level [46,52,57], surgery treatments [55,66], and chronic physical condition [60]. Other studies compared participants with and without diabetes [29,47,65], and fallers with non-fallers [54]. Additionally, several studies compared participants with LLA with able-bodied controls [29,47,63,64,67,109].

Of the interventional studies, 28 studies compared prosthetic components, including feet [73,78,79,82,88,93,95,99], knee systems [80,85,87,89,92,96], ankle systems [68,70,90], ankle-knee systems [69], liners [71,72,101], suspension systems [86,94], pylons [100], socket [83,91], a prosthetic alignment adapter [76], and one study compared both pylons and knee systems [98]. The remaining 6 articles were lifestyle intervention studies, targeting physical activity [75,81], walking capacity [84], rehabilitation [74], mobility and aerobic exercise training [97], and physical activity and weight management [77]. Only one interventional study investigated the relationship between real-world measurements and other variables, i.e., performance-based measures, self-reported outcomes, metabolic cost [70]. The 12 studies classified as algorithm/method development studies, aimed to develop and/or implement algorithms/models to classify activities [102,103,104,111], monitor and categorize the load regime [102,111,112,113], to combine GPS information with ambulatory activity data [105,110], to quantify cadence variability [108], and to incorporate doffing and donning information [109]. Two other studies developed a sensor design for monitoring limb-socket displacements [106,107]. The majority of the six validity/feasibility studies investigated the accuracy or validity of commercial activity monitors [114,115,117,119] or mobile phones [118]. One study tested the feasibility of a computerized algorithm to rate participants’ K-level [116]. Table 1 gives an overview of each article included in the review, structured according to study design category. 

### 3.3. Review Findings

#### 3.3.1. Categories of Reported Outcome Measures

The reported outcome measures related to real-world ambulation were merged into eight categories: step count, fitness and intensity of activity, type of activity and body posture, commercial scores, prosthetic use and fit, gait quality, GPS, and accuracy.

##### Step Count

Seventy-two articles reported on step count. The majority reported the number of steps taken per day [4,24,25,26,27,28,29,30,31,32,36,37,39,40,42,44,46,47,49,50,51,53,55,56,58,59,61,62,64,65,68,69,70,71,72,73,75,77,78,79,81,82,84,86,89,90,91,93,95,96,97,98,99,101,105,110,114,117]. However, articles also reported the number of steps for other time units, such as per week [76,85,100], per weekend- and weekend-day [21,42,49,62,76,85,98,100], per month and season [58], or total steps during the observation period [21,35,52,54,57,59,94,102,112,113,115]. Step count was also reported in combination with other variables of walking activity, such as number of steps per intensity activity [49,90], per walking technique [59], per walking bout [24,105], per activity classification [102,112], or the maximal number of consecutive steps taken [33]. Some articles reported on step count related to location, such as number of steps taken at home and away from home [70,105,110], per community category [54,57], or the difference in steps between a day in-patient and a day out-patient [36]. Step counts were also reported to demonstrate a difference between participant groups [4,26,29,31,32,37,40,46,47,49,51,52,54,55,56,64,65,68,75,77,81,84,105,110], between different types of activity monitors [115], or between baseline and follow-up measurements [24,40,50,68,77,81,84,89,93,96,97].

##### Fitness and Intensity of Activity

Thirty articles reported outcome measures that were related to a person’s fitness level or the intensity of the measured activity. The majority of the studies measured the cadence in steps per minute, which is an indication of ambulatory intensity [120]. Sixteen articles reported the time, frequency, or number of steps in specific intensity intervals, based on the parameters cadence or acceleration [4,21,27,28,45,47,49,51,61,62,81,90,95,101,117,118]. The number of intensity intervals and cut-off values were diverse, although most studies used the intensity intervals for low, medium, and high intensity activity. Some studies additionally included the time or frequency spent sedentary, i.e., inactivity [21,28,45,47,81,101]. Four articles demonstrated cadence distribution, by visualizing cadence per walking bouts categorized according to duration and number of bouts [86], or by quantifying the cadence variability [34,69,108]. Five articles reported the most intensive walking activity, by reporting the maximum or peak values of cadence averaged for a certain time-frame, such as the average cadence of the most intensive 60 min, 30 min, or 1 min [28,51,61,108,114]. Parameters related to activity intensity and fitness other than cadence, were walking speed [70,105], heart rate [63,67], or the acceleration of body movements in m/s^2^ or g (=9.81 m/s^2^) units [62,63,67,119]. Two articles reported the cadence variability scale parameter, which was a calculation of the distribution spread of cadence variability over the duration of the observation period [34,108].

##### Type of Activity and Body Posture

Twenty-six articles reported outcome measures that were related to the type of activity or body posture. To what extent the activity was specified varied among studies. Ten articles reported only the amount of activity and/or inactivity in duration, percentage or number of bouts, without further specifying for the type of activity [38,60,61,77,83,92,93,95,98,113]. Articles that specified the type of activity, reported activities such as stepping, walking, sitting, lying, standing, or other activities [24,37,41,43,62,63,66,67,68,80,91,109,119]. Two articles included an additional specification by classifying walking activity into different categories, such as turns [102] or directional locomotion [112]. Three articles reported the number of sit-to-stand transitions [26,63,67].

##### Commercial Scores

Fifteen articles reported a commercial score, or a score based on a custom calculation. Three articles reported a commercial score of the K-level [87,88,114]. Three articles reported a commercial score that indicated level of physical activity, i.e., the physical activity index [62], modus index [69], ambulation energy index [69], peak performance index [69], and Fitbit activity score [117]. The latter article also reported the Fitbit Web derived miles walked, calories, and number of floors climbed. However, since the commercial score did not account for height, weight and age of the users, the authors also developed a custom model for calculation of calories and the activity score including these variables [117]. Six articles reported a custom calculated score. Two articles calculated the K-level, using the three variables potential to ambulate, cadence variability and energy expenditure [48,116]. In addition, Orendurff et al. [116] reported a clinically judged K-level by a prosthetist, who subjectively rated the three variables in figures from the StepWatch data. Three articles reported the distance walked, of which two calculated distances using the clinically assessed step length multiplied by daily step count [33,96]. Darter et al. [97] reported distance walked and walking speed, but did not further describe the calculation.

##### Prosthetic Use and Fit

Eleven articles reported outcome measures related to prosthetic use and prosthetic fit. Five articles reported results on the duration that the prosthesis was worn, although using diverse terms [26,27,39,41,101]. Three studies included counts of doffing the prosthesis [41,109], or the duration of the prosthesis doffed [83]. Outcome measures related to prosthetic fit aimed to monitor displacement of the socket to the limb after wearing the prosthesis, and were measured through sensor pressure change or sensor signal loss during the wearing period [106,107].

##### Gait Quality

Seven studies reported outcome measures that were related to gait quality. Davis-Wilson et al. [29] assessed cumulative loading during ambulation that was calculated by the formula: daily steps/2 × peak ground reaction force, the latter was measured with a force plate and normalized to body weight. Kim et al. [105] calculated stride length from the three-dimensional position of the foot using IMU data. Frossard et al. [112,113] reported in two articles temporospatial parameters, i.e., the duration of the gait cycle, swing and support phases, and kinetic parameters, i.e., the forces, moments and impulses along the anteroposterior, mediolateral, and long axis of the prosthesis to categorize ADL activities into different locomotory activities. Two articles developed an algorithm to indicate gait quality, i.e., Kaufman et al. [80] calculated so-called gait entropy, and Gaunaurd et al. [74] developed a Machine Learning Classifier that gave biofeedback related to balance, toe load and knee flexion. Kaluf et al. [69] reported stance/swing time, that was calculated with the ModusTrex software.

##### GPS

Six studies included GPS data in addition to ambulatory activity measurements, however only five reported GPS-related results. Jamieson et al. [103] used GPS to record elevation data in the Strava app (Strava Inc., San Francisco, CA, USA), to aid with labelling uphill and downhill movement, but did not report data directly related to GPS data. Kim et al. [70,105] used a GPS-enabled smartphone in two studies in which non-sedentary periods were identified from the raw data and combined with the location to determine where the activity occurred. Results of daily steps, cadence and walking speed were divided into measures at home and away from home. Godfrey et al. [114] used GPS data to confirm whether steps were taken in the home or in the community to calculate the Modified Clinical K-level. In two other studies from Hordacre et al. [54,57], a GPS travel recorder was combined with a StepWatch to specify community activity into seven categories: employment, residential, commercial, health services, recreational, social, and home.

##### Accuracy

Four articles reported outcome measures of accuracy that were directly related to the wearable technology used. Griffiths et al. [104] reported F-scores and confusion matrices for eight different models that classified the postures sitting, standing, stepping and lying. Accordingly, Jamieson implemented classifiers and a neural network for activity recognition using eight models and three levels of label resolution [103]. They reported classification accuracy, F1-scores, and confusion matrices for the two models with the highest accuracy and the accuracy of the models for each participant with LLA [103]. Redfield et al. reported on the agreement between activity classification using one or two accelerometers [111], and van Dam et al. [119] reported the test-retest reliability of activity monitor by identical assessments on two separate days.

#### 3.3.2. Reported Outcome Measure in Categories Per Study Design

The most frequent reported outcome measure category was step count, followed by outcome measures related to fitness/intensity of activity (Table 2).

## 4. Discussion

The overall purpose of this scoping review was to survey the scientific literature to evaluate the use of wearable activity monitors in reporting real-world ambulation and prosthetic use in people with LLA. The results demonstrate that the number of studies using wearable technologies is rising, hence it is important to understand the opportunities and limitations in the use of these devices. By classifying the included articles according to their study design, we demonstrated that the number of algorithm/method development studies and validity/feasibility studies was relatively low, most likely because these studies are challenging to perform in the real world and hence are more often conducted in the laboratory [10]. The majority of the studies using wearable technologies in the real world were observational and interventional studies. This is not surprising, as wearable technologies enable monitoring a person’s natural behavior and allow for observation over time, or assessment to the effect of an intervention. Although there exists a large battery of performance-based tests that can detect changes in physical functions and capacity [121], real-world measurements have revealed that capacity is not necessarily the same as performance [34,37]. Studies have shown that half of older community dwelling adults classified by clinic-based tests as high functional capacity, exhibit low functional level behavior in the community [48]. The use of wearable technology in the real world extends the understanding of a person’s natural behavior by monitoring parameters that have not been feasible to perform in-laboratory.

This review identified multiple outcome measures that are available to monitor since the use of wearable technologies in the real world. The most frequent reported outcome measure was daily step count, which is an indication of the level of physical activity. Results of multiple studies have demonstrated that the majority of people with LLA do not meet the recommended level of physical activity [16]. Lower levels of physical activity in this population is associated with an increased risk of developing cardiovascular diseases [122], and lower perceived quality of life [123]. Monitoring physical activity may facilitate the development of personalized treatments that optimize the individual health status. Step count is also used to calculate cadence (steps min^−1^), a measure that is closely related to walking speed and hence, indicates the intensity of walking [124]. Walking intensity classified in intervals provides valuable information about the structure of daily walking activity. For instance, Kim et al. [105] showed that people with LLA and able-bodied control persons had a similar variance in walking intensity, but the LLA group had a more positively skewed distribution of intensity, indicating that both groups had similar ranges of intensity, but that the LLA group took more of their steps at lower cadence. The cadence variability, i.e., the ability to walk at multiple speeds, is considered an important determinant of functional mobility and hence community ambulation. Some studies used cadence to report the upper boundaries of physical activity [28,46,51,61,108,114]. Peak values of performed physical activity are an indication of a person’s fitness and ability to perform high-intensity physical activity. This is important, because previous research demonstrated that a larger amount of high intensity physical activity is associated to higher cardiorespiratory fitness [28]. Despite the scarce evidence on this topic, the results indicate that assessment of the most intensive physical activity performed in the real world can be a valuable measure to assess overall health status.

Other identified outcome measures in this review were related to prosthetic use and information about the environment in which the prosthesis is used. The amount and structure of prosthetic use is directly related to the amount of prosthetic ambulation, which again, is an indication of prosthetic fit and trust in the prosthesis. For instance, studies have shown that donning and doffing the prosthesis influences limb fluid volume, and temporarily doffing the socket is necessary to facilitate limb fluid volume recovery that is retained during subsequent activity [107,109]. However, Balkman et al. [41] argue that frequent donning and doffing of the prosthesis can be an indication of a poorly fitted prosthesis, that can cause skin problems on the stump. Monitoring when, how and how much a person uses the prosthesis can provide valuable information to clinicians about prosthetic fit and functioning. To investigate the amount and location of prosthetic use in the community, multiple studies have used GPS and found that ambulatory patterns outside the home are different from inside the home [70,105]. Ambulation away from home requires a higher level of functional mobility, because it generally covers larger distances, and is influenced by environmental factors such as obstacles, terrain, and variable weather conditions [54,57]. Jamieson et al. [103] used recordings from a chest-mounted camera in addition to GPS data to determine the type of terrain that participants walked over. They observed variation among the participants, i.e., some participants walked on certain terrains that other participants avoided or rarely walked over, such as sandy terrain [103]. Hence, assessment of the amount and patterns of community activity is important in prosthesis prescription and to examine the ability for participation in the society, which is an important determinant of quality of life [54].

By categorizing the identified outcome measures in this review, we were able to obtain a clear overview of which categories were reported in different study design. The results showed that of the eight categories, the category step count was the most frequent reported category, regardless of the study design. It is, however, arguable whether step count is an appropriate outcome measure for different research questions. To obtain a sufficient degree of construct validity in a study, it is important that the chosen outcome measures reasonably represent what it intends to measure. A surprising observation in this review is that 82% of the interventional studies report one or multiple outcome measures related to step count, however, the majority of the studies did not find a significant effect of the intervention on daily step count [70,71,72,73,76,78,82,85,86,90,91,93,95,96,98,99,100]. For instance, Klute et al. [116] measured improvements in kinematic and metabolic walking efficiency in laboratory tests using a microprocessor knee versus a hydraulic knee, but reported no change in real-world ambulatory patterns. Accordingly, Andrysek et al. [85] showed no difference in step count between an automatic stance-phase lock knee (ASPL) and a weight-activated braking knee, despite the lower energy expenditure measured for the ASPL knee. Moreover, participants in this study rated the ASPL knee higher in terms of knee stability and improved walking, which could be interpreted as encouraging factors for prosthetic use, but this did not result in increased step count. Segal et al. [90] demonstrated that participants wearing a torsion adapter tended to take more low- and medium-intensity steps, but fewer high-intensity steps compared to a rigid adapter. However, total daily step count was not different between the adapters, indicating that the structure of walking might change, but not the total amount of ambulation [90]. According to Wurdeman et al. [82], changing a prosthesis will change the biomechanics of the individual, such as the step length, but not the behavior and daily routines that mainly determine the number of steps walked. Yet, interventions targeting behavior change, such as physical activity level, have neither demonstrated significant increases in daily step count [75,77]. A few studies have shown small increases immediately after the intervention, but these effects disappeared on the long-term [81,84]. Imam et al. [84] demonstrated a long-term improvement in walking capacity, but the intervention did not result in participants increasing their physical activity level. Hence, it is suggested that it is the individual’s willingness to change or changes in daily routines that can lead to behavior change, rather than any enabling technology or intervention [101]. Likewise, observational studies that aim to gain understanding of mobility are sometimes limited in construct validity. Anderson et al. [31] found no difference in daily step count between fallers and non-fallers, indicating that number of steps is not necessarily a determinant for falls. The participants experienced falls mostly caused by intrinsic destabilization sources, inadequate weight shift patterns, and transfer-related functional activities, i.e., factors that are related to balance, and it would therefore be more likely to detect a between-group difference in parameters related to balance [31]. Overall, our findings demonstrate the often-used outcome measure step count has limited ability to detect changes in walking behavior, and this might have consequences on a study’s construct validity. Hence, researchers should consider whether they capture the relevant information when designing their studies. The challenge whether sampled information is representative of the investigated situation was earlier demonstrated in the representative design developed by Brunswik [125]. The representative design, which is a methodological approach to achieve generalizability of results, requires researchers to sample information that is representative of the ‘target ecology’, and to specify how those conditions are represented in the experiment. Building on this approach, we encourage researchers to define what they are interested in to measure in their experiments and reflect on whether the selected outcome measure might answer their research question. Additionally, we recommend researchers to elaborate more precisely on the limitations of the reported outcome measures to avoid misinterpretation of the results.

As discussed earlier, in-laboratory studies of the mobility of people with LLA are often limited in the extent that study findings can be generalized to real-world situations [5]. Measurements in the real world may overcome some of the limitations of in-laboratory testing, enhancing the ecological validity of the studies. According to the definition of Martin T. Orne, ecological validity refers to the generalization of experimental findings to the real world outside the laboratory [126]. Despite the increasing popularity of studying people with LLA outside the laboratory, ecological validity is a rarely discussed topic in the field of prosthetic mobility [127]. Among the included studies in this review, only four studies mentioned the term ecological validity, without specifying what the terms implies and how it is relevant with regards to the interpretation of their results [59,73,78,112]. This concern was earlier expressed by Holleman et al. [128] in the field of social sciences where there is an ongoing debate about the definition of ecological validity and how to enhance the understanding of human behavior in the real world. There seems to be no agreement upon a definition in the literature, nor any form of classification or quantitative approach to determine or evaluate a study’s ecological validity. Holleman et al. [128] describe that technological advances have further stimulated researchers to emphasize the importance of studying human behavior in the real world. However, they additionally argue that labeling an experiment as ‘ecological valid’ because it is conducted in a ‘real-world’ environment can lead to misleading and potentially counterproductive conclusions [128]. Therefore, they highlight the importance of developing and criticizing the contemporary framework of ecological validity. The contemporary framework for evaluation of ecological validity includes the dimensions stimuli, tasks, behaviors, and research context that can be evaluated on a continuum of artificiality versus naturality and simplicity versus complexity. Whereas the in-clinic environment is characterized by its artificiality and simplicity, the real world is at the other extreme and is characterized by its naturality and complexity. With respect to the included studies in this review that monitor prosthetic ambulation in the real world, the environment is in principle higher in ecological validity compared to laboratory studies. Namely, the study subjects perform their normal behavior, without receiving instructions and without any other demand characteristics that can influence their behavior. However, based on the results of this review, we believe that the contemporary wearable technologies are limited in the ability to capture the essential information of real-world ambulation. This review demonstrated a poor diversity of reporting outcome measures, in particular studies using commercial devices rather than custom-developed devices were limited to reporting step count or the intensity of activity. Therefore, we wish to introduce some suggestions to the future development of wearable technologies. First, we observed that few studies included essential determinants of community ambulation, such as parameters related to prosthetic fit or gait quality (11% and 7% of the included articles, respectively). Research has demonstrated that gait symmetry and step length, i.e., indications of gait quality, are associated with performance-based measures [129], and that walking capacity is associated with walking performance in the community [28,61]. Hence, gait quality might also be associated with the amount and structure of community ambulation. Therefore, technological development and advancement of wearable sensors should include outcome measures of gait quality of prosthetic ambulation. Second, we observed that parameters related to balance, which is an important determinant of prosthetic mobility [130], are not yet included in the features of the contemporary wearable technology, though several studies have demonstrated that balance confidence is associated with the level of community activity and participation [42,49]. Hence, future research should investigate the potentials of including parameters related to balance in advancement of wearable technologies. Last, mobility involves dimensions that are challenging to quantify, such as pain, fear, motivation, confidence, or other psychosocial aspects [3,32]. Enhancing the understanding of the complexity of prosthetic mobility in daily life may facilitate further development of wearable technologies for the purpose of monitoring ambulatory behavior in this population. As such, we recommend future researchers to utilize studies that investigate prosthetic mobility in daily life using a holistic approach, such as performed by e.g., Hafner et al. [8], Batten et al. [3], and Miller et al. [32]. On the other hand, we recognize that the complexity of real-world ambulation and diversity in human behavior might go beyond the potentials of technology. Yet, technological advancements that aims to integrate more variables that are important determinants of prosthetic mobility can enhance the opportunities to capture essential information of real-world ambulation.

### Limitations

First, our review was limited to English publications only, and may have excluded important studies published in other languages. Second, the classification of articles based on study design was performed by the two reviewers and judged according to the aim of the study. Many studies had multiple study objectives that could be considered under different study designs, such as interventional studies that in addition had objectives that were essentially observational, or algorithm/development studies that also included a form of accuracy assessment. However, we believe the classification of study designs used in the present review, is appropriate for describing the main objective of each study. Third, the synthesis of outcome measures categories was a subjective evaluation by the reviewers. The large variety in outcome measures and related units may have caused somewhat overlap between categories. For instance, cadence was considered as an indication of walking intensity, although it is in essence based on the number of steps. However, we believed that the category step count was more related to the level of physical activity, while the intensity of activity is more related to the structure of walking throughout the day. Last, our concern regarding the extent that reported outcome measures can answer the study’s research question was based on an overall evaluation of all studies included in this review. To judge each individual study goes beyond the scope of a scoping review [128]. Yet, we believe that our evaluation was sufficient to emphasize the need to report outcome measures that capture the essential information of real-world ambulation of people with LLA.

## 5. Conclusions

To the best of our knowledge, this is the first review that presents the reported outcome measures in studies of real-world ambulation in people with LLA. We identified that the most frequent used outcome measure was related to step count, regardless of study design. We have expressed our concerns that step count might not be a reliable outcome measure to detect change of an intervention, as step count is highly dependent on a person’s daily routine. Other important outcome measures were less reported, such as outcome measures related to the type of activity, or the intensity of activity. Only few studies reported outcome measures related to gait quality or prosthetic fit. In future research, we encourage researchers to reflect on whether the selected outcome measures are representative of the investigated situation, and to elaborate on the limitations of the reported outcome measures. Additionally, we argue that the contemporary technology is limited in providing a comprehensive picture of real-world ambulation. In future development of wearable technologies, we encourage researchers to integrate variables that are important determinants of prosthetic mobility. Furthermore, as the use of wearable technology in the real world is expected to further increase, we encourage researchers in the rehabilitation sciences to engage in the debate and development of the definition and framework of ecological validity.

## Figures and Tables

**Figure 1 sensors-22-02243-f001:**
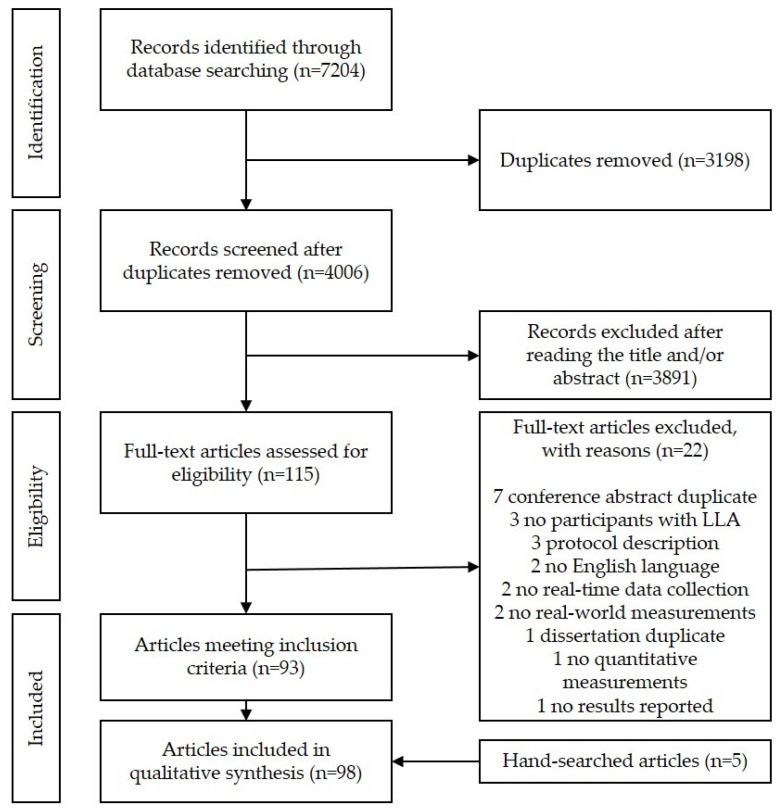
PRISMA flowchart for scoping review outcome measures used in studies to real-world ambulation in people with a lower limb amputation.

**Figure 2 sensors-22-02243-f002:**
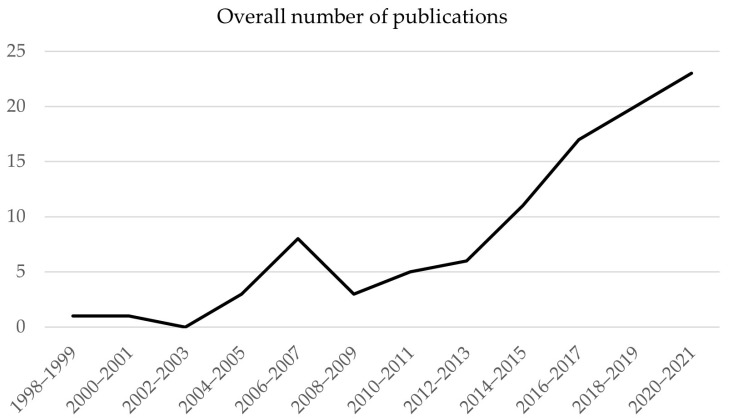
Number of publications grouped into 2–year bins.

**Table 1 sensors-22-02243-t001:** Overview of each article included in the review, structured according to study design category.

No.	First Author (Year, Country)	Title	Objective(s)	Study Population (*n* (Female)); Mean ± SD Age; Level (*n*); Etiology (*n*)	Technology; Placement; Duration	Reported Outcome Measures	Key Findings
**Observational studies**							
1	Tolani (2021, USA) [24]	Understanding changes in physical activity among lower limb prosthesis users: A COVID-19 case series (clinical letter)	Understand potential changes to Physical Activity (PA) during shutdown and “shelter-in-place” orders.	*n* = 4(0); between 20–79; 2 Trans-Tibial Amputation (TTA), 1 Trans-Femoral Amputation (TFA), 1 Knee Disarticulation (KD); 3 non-dysvascular, 1 dysvascular/diabetic	EmpowerGO; prosthesis; between 74 and 200 days	Steps per day (overall, pre-index, post-index); supplemental data: number of bouts; steps per bout; time per bout; steps per day normalized to pre-index step count	Two participants demonstrated clear signs of overall reduced activity through beginning stages of the COVID-19 pandemic.
2	Rosenblatt (2021, USA) [25]	Prosthetic disuse leads to lower balance confidence in a long-term user of a transtibial prosthesis	Assess the impact of prosthesis disuse on balance, gait, PA and balance confidence.	*n* = 1(0); 76; TTA; cancer	StepWatch 3; prosthetis; 2 × 7 days	Steps per day	Balance confidence, walking speed and steps per day decreased with 19%, 12%, and 19%, respectively, following 4 months of prosthesis disuse; functional measures were not impacted.
3	Miller (2021, USA) [26]	Patterns of sitting, standing, and stepping after lower limb amputation	Describe sitting, standing, and stepping patterns and compare the patterns between people with dysvascular Lower Limb Amputation (LLA) and traumatic LLA.	*n* = 32(5); 62.6 ± 7.8; 22 TTA, 7 TFA/KD; 15 trauma, 17 dysvascular	ActivPAL; thigh; 10 days	Steps per day; wake time (min/day); number of sit-to-stand transitions; sitting, standing and stepping in categorized bout durations (min/day), proportion/day, bouts/day)	Participants spent most time sitting. PA bouts were mostly <1 min. Significant between-etiology differences for sitting and standing time.
4	Mellema (2021, Norway) [27]	Impact of the COVID-19 restrictions on physical activity and quality of life in adults with lower limb amputation	Investigate the impact of COVID-19 restrictions on ambulatory activity and Health-Related Quality of Life (HR-QoL).	*n* = 20(4); 56.2 ± 11.9; 12 TTA, 2 KD, 5 TFA, 1 bilateral TTA; 9 trauma, 4 cancer, 2 diabetes, 5 others	StepWatch 4; prosthetic ankle; 2 × 7 days	Steps per day; prosthetic wear time (hours/day); time in low, moderate, and high intensity level	Prosthetic wear time decreased significantly. Daily step count, moderate-intensity and high-intensity ambulation, and HR-QoL increased, but low-intensity ambulation decreased.
5	Mellema (2021, Norway) [28]	Relationship between level of daily activity and upper-body aerobic capacity in adults with a lower limb amputation	Investigate the relationship between upper-body peak aerobic capacity (VO2peak), PA levels, and walking capacity.	*n* = 14(2); 55.7 ± 10.1; 7 TTA, 2 KD, 5 TFA; 8 trauma, 2 cancer, 1 congenital, 3 others	StepWatch 4; prosthetic ankle; 7 days	Steps per day, time in sedentary, low, moderate, high intensity (%); peak intensity level	VO2peak correlated significantly with daily step count, sedentary time, high-intensity activity level, and peak-intensity activity level, preferred walking speed, and 2-min walking test.
6	Davis-Wilson (2021, USA) [29]	Cumulative loading in individuals with non-traumatic lower limb amputation, individuals with diabetes mellitus, and healthy individuals (conference abstract)	Determine if differences existed in cumulative loading between individuals with diabetes + LLA, individuals with diabetes, and healthy individuals of similar health.	*n* = 6(0); 58 ± 6; level and etiology of amputation N/A	ActiGraph GT3X; hip; 10 days	Steps per day; cumulative loading (body weight/day)	No differences in cumulative loading between diabetes + LLA and diabetes groups, but diabetes + LLA had lower cumulative loading compared to healthy individuals.
7	Chihuri (2021, USA) [30]	Quantify the risk of falls and injuries for amputees beyond annual fall rates—A longitudinal cohort analysis based on person-step exposure over time	Determine all-cause fall and injury rates over time, accounting for daily per person-step exposure.	*n* = 10(3); 48.7 ± 12.5; 7 TTA, 3 TFA; 6 Peripheral Artery Disease (PVD), 2 trauma, 2 non-chronic medical cause	StepWatch 4; prosthetic ankle; 5 × 1 week	Steps per day	Limited community walking ability was associated with higher incidence of falls and injuries when accounting for person-steps.
8	Anderson (2020, USA) [31]	Falls after dysvascular transtibial amputation: A secondary analysis of falling characteristics and reduced physical performance	Characterize falls using existing Fall-Type Classification Framework and describe functional characteristics across the framework categories.	*n* = 69(N/A); 64.5 ± 8.6; 64 TTA, 1 TFA, 1 KD, 1 bilateral TTA & TFA, 2 bilateral TTA; all dysvascular	ActiGraph GT3X-BT; waist belt; 10 days	Steps per day	43.5% of participants reported falls, of which the incidence was highest for intrinsic destabilization sources, from incorrect weight shift patterns during transfer activities.
9	Miller (2020, USA) [32]	Psychosocial factors influence physical activity after dysvascular amputation: A convergent mixed-methods study	Identify psychosocial factors with potential to influence clinically relevant measures of PA, physical function, and disability.	*n* = 20(2); 63.4 (57.5 70.0 interquartile range); 15 TTA, 2 TFA, 3 bilateral; all dysvascular	ActivPAL 3; thigh; 10 days	Steps per day	PA results from an interaction among perceptions of prosthesis, fear during mobility, influence of LLA on life activities, and positive outlook within social interactions.
10	Hofstad (2020, The Netherlands) [33]	Maximal walking distance in persons with a lower limb amputation (letter)	Assess the number of consecutive steps and walking bouts, using an accelerometer sensor.	*n* = 20(7); 68 (range 60–74); 9 TTA, 4 KD, 7 TFA; 6 trauma, 10 dysvascular, 2 cancer, 2 other	3 tri-axial piezo-capacitive MiniMods Dynaport; 2 on each side of trouser pocket, 1 on sternum; 2 days	Maximal consecutive steps; frequency per hour of number of steps per bin; maximal walking distance (meters)	The SIGAM mobility grade did not reflect what participants do in daily life. Objective assessment of maximal number of consecutive steps or maximal covered distance is recommended.
11	Beisheim (2020, USA) [34]	Performance-based outcome measures are associated with cadence variability during community ambulation among individuals with a transtibial amputation	Evaluate whether physical performance (10—meter Walk Test-based walking speeds, L-Test, and Figure-of-8 Walk Test scores) is associated with community-based cadence variability.	*n* = 41(15); 58.3 (range 54.6–62.0); all TTA; 24 dysvascular, 13 trauma, 1 cancer, 1 congenital, 2 other	FitBit One; ankle; 7 days	Cadence variability (Weibul probability density) (steps/min); cadence variability scale parameter	Beyond covariates, faster self-selected gait speed best predicted increased cadence variability during community ambulation.
12	Zhang (2019, US) [35]	Evaluation of gait variable change over time as transtibial amputees adapt to a new prosthesis foot	Investigate whether gait variables were affected by the duration of accommodation period, and assess relationship between measures outcomes and subjective perception.	*n* = 7(0); 53.0 ± 15.2; all TTA; etiology N/A	Up move; prosthesis; 5 days	Total steps (for each participant)	Significant changes in gait speed and double support time during early phase, but gait variables did not significantly change during day 2–5. Visual Analog Scale (VAS) scores correlated with step count and cadence.
13	Sherman (2019, UK) [36]	Daily step count of British military males with bilateral lower limb amputations: A comparison of in-patient rehabilitation with the consecutive leave period between admissions	Determine whether mean daily step count changed between in-patient rehabilitation and consecutive leave periods.	*n* = 9(0); 26.0 ± 6.0; all bilateral TTA/TFA/Trans-Humeral (THA)/Trans-Radial (TRA)/KD; all trauma	Long-Term Activity Monitor (LAM2); prosthesis; 2 × 2 weeks	Steps per day	Step count decreased when away from rehabilitation.
14	Pepin (2019, USA) [37]	Correlation between functional ability and physical activity in individuals with transtibial amputations: A cross-sectional study	Investigate association between functional ability and PA.	*n* = 19(4); 59.6 ± 10.8; all TTA; 2 trauma, 17 non-traumatic	ActivPAL; thigh; 7 days	Steps per day; duration lying/sitting, standing (hours); duration stepping, walking (minutes)	Number of steps per day had a moderate to good correlation with the Amputee Mobility Predictor (AMP) and a fair correlation with the Timed Up and Go (TUG) and 2 Minute Walk Test (2MWT).
15	Musig (2019, Germany) [38]	Relation between the amount of daily activity and gait quality in transfemoral amputees	Examine kinematic variability during walking and the association with daily activity.	*n* = 15(1); 44.0 ± 9.0; 11 TFA, 4 KD; etiology N/A	VitaMove (Activ 8), prosthesis stem, 7 days	PA per day (min/day)	Significant correlation between daily activity and variability in the trunk and pelvis, and gait velocity.
16	Miller (2019, USA) [39]	Physical function and pre-amputation characteristics explain daily step count after dysvascular amputation	Identify factors that contribute to daily step count.	*n* = 58(3); 64.4 ± 9.0; 55 TTA, 3 other; all dysvascular	ActiGraph GT3X-BT; waist; 10 days	Steps per day; prosthetic wear time (minutes)	Physical function, cardiovascular disease, and pre-amputation walking time explained 62% of daily step count.
17	Klute (2019, USA) [40]	Daily step counts and use of activity monitors by individuals with lower-limb loss (conference abstract)	Observe and determine willingness to use smart activity monitors in daily life, and discover if self-monitoring increases PA levels.	*n* = 74(N/A); 52.0 ± 15.0; 56 TTA, 11 TFA, 1 KD; 42 trauma, 14 dysvascular, 18 other	Fitbit Zip; placement N/A; multi-year period (no specific duration reported)	Steps per day; habitual device use (%)	Self-monitoring of activity levels did not result in higher activity. Participants demonstrated habitual use approximately one quarter of the time.
18	Balkman (2019, USA) [41]	Prosthetists’ perceptions of information obtained from a lower limb prosthesis monitoring system: a pilot study	Assess prosthetists’ perceptions of prosthesis use and activity information obtained by a monitoring system.	*n* = 3(1); 50.1 ± 22.7; all TTA; 1 trauma, 2 dysvascular	Proximity sensor (WAFER) and 2 ActiGraph GT3X+; socket, thigh and ankle; 2 weeks	Prosthesis use (hours/day); time sitting, standing, walking (hours/day); times doffing prosthesis (for each participant)	Prosthetists over- and under-estimated patient activity, relative to monitored activity, and found features of multiple report formats clinically useful.
19	Sions (2018, USA) [42]	Self-reported functional mobility, balance confidence, and prosthetic use are associated with daily step counts among individuals with a unilateral transtibial amputation	Determine if functional mobility, balance confidence, and prosthetic use are associated with PA.	*n* = 47(16); 58.5 ± 12.0; all TTA; 20 infection, 16 trauma, 5 dysvascular, 2 cancer	StepWatch; prosthesis; 7 days	Steps per day	Self-reported functional mobility and balance confidence each explained 13% of the variance in step count, whereas prosthetic use explained 10%.
20	Sanders (2018, USA) [43]	Residual limb fluid volume change and volume accommodation: Relationships to activity and self-report outcomes in people with trans-tibial amputation	Examine how activities and self-report outcomes relate to daily changes in residual limb fluid volume and volume accommodation.	*n* = 29(5); 56.7 ± 14.8; all TTA; 20 trauma, 7 dysvascular, 2 congenital	ActiGraph GT3X-BT; prosthesis; 3 h	Time sitting, walking, standing, weight-bearing (sum of standing and walking), and prosthesis doffed (%)	Morning-to-afternoon percent limb fluid volume change per hour was not strongly correlated to percent time weight-bearing or to self-report outcomes.
21	Esposito (2018, USA) [44]	Daily step counts in Service Members with lower limb amputation (conference abstract)	Quantify PA in the months following amputation.	*n* = 27(N/A); 16 TTA, 7 TFA, 1 bilateral TTA, 3 bilateral TFA; etiology N/A	StepWatch 3; placement N/A; 7 days	Steps per day	Participants walked 3.142 ± 1.308 steps per day. No indications that step count increased farther along in the rehabilitative process.
22	Samuelsen (2017, USA) [45]	The impact of the immediate postoperative prosthesis on patient mobility and quality of life after transtibial amputation	Examine activity level and quality of life for patients receiving an immediate postoperative prosthesis.	*n* = 10(1); 58 (range 22–69); all TTA; all PVD	ActiGraph GT3X-BT; waist; 6 weeks	Cadence; time in sedentary, light activity, and moderate to vigorous activity (cutoff values 0–99, 100–2019, 2020–5998, and >5999, respectively) (%)	Participants spent 88% of their time sedentary, 11.5% in light, and 0.3% in moderate to vigorous activity level, and had low physical and emotional scores.
23	Juszczak (2017, USA) [46]	Developing an evidence based approach to address functional level changes in persons following amputation (conference abstract)	Incorporate ambulatory activity monitors (SAM) to collect objective functional mobility data to assess functional improvements during the rehabilitation and to improve prosthetic prescription.	*n* = 10(N/A); 53.2 ± 13.4; all TFA; etiology N/A	StepWatch; placement N/A; duration N/A	Steps per day; time moderate/intense level of ambulation (%)	Patients with higher K-level classification ambulated to a greater capacity, higher intensity, and for a sustained period of time compared to lower K-level. SAM may be effective for evaluating functional level change and assessing prosthetic needs.
24	Paxton (2017, USA) [47]	Physical activity, ambulation, and comorbidities in people with diabetes and lower-limb amputation	Characterize PA and its relation to physical function and comorbidities for diabetes and transtibial amputation (DM + AMP), diabetes without AMP (DM), and nondisabled adults.	*n* = 46(7) (22 with AMP); 62.3 ± 10.3; level N/A; all (22) dysvascular	ActiGraph GT3X-BT; waist; 10 days	Steps per day; time in sedentary, light, moderate, vigorous and very vigorous intensity (%)	Nondisabled group had more PA than DM, who performed more than DM + AMP. PA was related to physical function in DM and DM + AMP, but not to number of comorbidities.
25	Orendurff (2016,USA) [48]	Functional level assessment of individuals with transtibial limb loss: Evaluation in the clinical setting versus objective community ambulatory activity	Determine relationship between K-level determined in the clinic and K-level based on real world ambulatory activity.	*n* = 12(1); 57 ± 12; all TTA; etiology N/A	StepWatch; prosthesis; 7 days	Calculated K-level	Good agreement between the two methods of determining K-level. Clinic-based ambulatory capacity correlated with real-world ambulatory behavior.
26	Mandel (2016, Canada) [49]	Balance confidence and activity of community-dwelling patients with transtibial amputation	Examine relationship between balance confidence and community-based PA	*n* = 22(8); 61.4 ± 7.6; all TTA; 14 dysvascular/diabetes, 6 trauma, 2 cancer	StepWatch; prosthesis; 7 days	Steps per day; steps in frequency categories low (<16 steps/min), medium (16–40 steps/min), high (>40 steps/min)	Balance confidence was significantly lower among subjects with <3.000 steps/day. Balance confidence was significantly correlated with total steps.
27	Desveaux (2016, Canada) [50]	Physical activity in adults with diabetes following prosthetic rehabilitation	Determine if adults with diabetes and TTA meet PA guidelines, if PA is maintained post-rehabilitation and if physical functions are associated with PA.	*n* = 15(5); 61 ± 12; all TTA, all dysvascular	StepWatch; ankle intact limb; 9 days	Steps per day; weekly minutes of Moderate to Vigorous PA (>90 steps/min) (MVPA)	Participants took 3809 ± 2189 steps per day and 24 ± 41 weekly minutes of MVPA, below the guidelines. Outcomes remained stable post-rehabilitation. PA was correlated to 2MWT and gait speed.
28	Chu (2016, Hong Kong) [51]	Comparison of prosthetic outcomes between adolescent transtibial and transfemoral amputees after Sichuan earthquake using Step Activity Monitor and Prosthesis Evaluation Questionnaire	Investigate daily step activities and prosthesis-related quality of life amputees after the earthquake.	*n* = 21(5); 14.6 ± 2.3; 11 TTA, 10 TFA; all trauma	StepWatch; prosthesis; 3 months	Steps per day; duration low (<15 steps/min), medium (15–40 steps/min), high (>40 steps/min) (hours); peak activity index; endurance score; cardiovascular score; peak 5-min burst, peak 1–min burst	TTA had significantly higher step activity than TFA (4577 ± 849, 2551 ± 693, respectively). All participants showed daily wearing time > 12 h/day. Prosthesis Evaluation Questionnaire (PEQ) was not different between-groups.
29	Arch (2016, USA) [52]	Real-world walking performance of individuals with lower-limb amputation classified as Medicare functional Classification level 2 and 3	Investigate outcomes of in-clinic performance-based evaluations and real-world walking performance measures.	*n* = 27(6); 56.8 ± 12.2; 20 TTA, 7 TFA, etiology N/A	Fitbit One; prosthetic ankle; 7 days	Total steps; total activity (minutes); activity in low (1–30 steps/min), moderate (>30–60 steps/min), high (>60 steps/min) activity (%)	K2 had significantly slower walking speed, shorter distance walked in 6 min, total step count and fewer active minutes than K3.
30	Kent (2015, USA) [53]	Step activity and stride-to-stride fluctuations are negatively correlated in individuals with transtibial amputation	Determine if increased stride-to-stride fluctuations correspond to a reduced level of activity.	*n* = 22(N/A); 52.0 ± 10.9; all TTA; 13 trauma, 5 diabetes, 2 dysvascular, 1 cancer, 1 infection	ActiGraph; pylon; 3 weeks	Steps per day	Increased stride-to-stride fluctuations were related to decreased activity levels.
31	Hordacre (2015, Australia) [54]	Community activity and participation are reduced in transtibial amputee fallers: A wearable technology study	Use wearable technology to assess activity and participation characteristics in the home and various community settings for fallers and non-fallers.	*n* = 47(11); 59.7 (range 19–98); all TTA; 18 PVD, 17 trauma, 11 other	StepWatch 3, QStarz BT-Q1000XT Global Positioning System (GPS); prosthesis; 7 days	Step count, number of visits, total steps and visits per community categories (employment, residential, commercial, health service, recreational, social, other); total steps at home	Fallers had significantly lower community activity levels and participation than non-fallers, specifically for recreational and commercial roles.
32	Parry (2014, USA) [55]	Gait outcome of pediatric lower extremity amputation patients with and without skin grafts (conference abstract)	Test hypothesis that lower extremity amputees with skin grafts on the amputation site had poorer function than those without skin grafts.	*n* = 13(N/A); 13.5 ± 4.6; level N/A, all trauma	Step Activity Monitor; placement N/A; 3 days	Steps per day	The two groups demonstrate comparable gait quality, gait efficiency, prosthetic use and self-reported functional ability.
33	Lin (2014, USA) [56]	Physical activity, functional capacity, and step variability during walking in people with lower-limb amputation	Explore relationship between PA and 6 Minute Walk Test (6MWT), step length variability, step width variability and Preferred Walking Speed (PWS)	*n* = 20(5); 50.6 ± 10.6; 12 TTA, 7 TFA, 1 KD; 12 trauma, 7 dysvascular, 1 other	Impulse model B-1 Pedometer; waist; 7 days	Steps per day	PA correlated strongly to PWS, 6MWT, and fairly to step width variability, but was inversely related to step length variability of both legs.
34	Hordacre (2014, Australia) [57]	Use of an activity monitor and GPS device to assess community activity and participation in transtibial amputees.	Assess ability to use wearable technology to measure community activity and participation, and determine if community activity and participation was different for predicted K-levels.	*n* = 46(N/A); 64.7 ± 13.8; all TTA; 19 trauma, 18 PVD, 9 other	StepWatch 3 and QStarz BT-Q1000XT GPS; prosthesis; 7 days	Step count, number of visits, total steps and visits per community categories (employment, residential, commercial, health service, recreational, social, other, home, lost in linkage, unidentified); total steps at home; community step count and visits per K-level (K1/2, K3, K4)	Participants completed on average 16.645 community steps and 16 visits over seven days. K1 and K2 had significant lower levels of community activity and participation than K3 and K4.
35	Halsne (2013, USA) [58]	Long-term activity in and among persons with transfemoral amputation	Study habitual activity in free-living environments, and explore relationships between Medicare Functional Classification Levels (MFCL) and performance.	*n* = 17(4); 49.1 ± 16.4; all TFA; 10 trauma, 3 malignancy, 1 dysfunction, 1 vascular, 1 infection	StepWatch; prosthesis; 12 months	Steps per day (for each participant); steps per day (sample mean); change in step count per season and per month (%)	Subjects took 1.540 steps per day, and activity increased with MFCL. Warmer seasons and months promoted higher activity, but peak temperatures and humidity depressed activity.
36	Highsmith (2012, USA) [59]	Spatiotemporal parameters and step activity of a specialized stepping pattern used by a transtibial amputee during a Denali mountaineering expedition	Describe spatiotemporal differences between the specialized French technique and traditional stepping and report step activity during a climbing expedition in Denali, AK, USA.	*n* = 1(0); 51; TTA; trauma	Sportline ThinQ XA Model 305 Pedometer; on a lanyard around the neck; 8 days	Steps per day; total step count; steps per technique (*n*, %)	The French technique had higher stride, step, and double support times than traditional stepping, but lower velocity and stride and step lengths. 27% of the steps were taken using the French technique.
37	Van den Berg-Emons (2010, The Netherlands) [60]	Accelerometry-based activity spectrum in persons with chronic physical conditions	Give an overview on the impact of chronic physical conditions on everyday PA and identify high-risk conditions, and compare objective activity levels with the levels estimated by rehabilitation physicians.	*n* = 18(1); 56 ± 13; all bilateral TTA; 9 trauma, 9 vascular	5 ADXL202 uniaxial piezoresistive accelerometers; 2 thigh, 2 sternum, 1 wrist; 48 h	Duration physical activities (% of 24h-day); proportion of physical activities of able-bodied subjects (%)	Lowest activity levels were among vascular TTA, spinal cord injury, and myelomeningocele, less than 40% of the able-bodied level. Rehabilitation physicians considerably underestimated the magnitude of inactivity.
38	Parker (2010, Canada) [61]	Ambulation of people with lower-limb amputations: relationships between capacity and performance measures	Examine relationship between ambulation capacity and community performance, and explore what demographic and clinical variables influence ambulation performance.	*n* = 52(11); 55.2 ± 4.5; 30 TTA, 16 TFA, 6 bilateral TTA; 26 trauma, 20 vascular, 6 other	StepWatch 3; prosthetic ankle; 7 days	Steps per day; activity per day (minutes), time in low (1–30 steps/min), medium (>30–60 steps/min), high (>60 steps/min) activity (%); peak activity index (mean of highest 30 min steps/min)	2MWT was significantly related to step activity measures and Trinity Amputation and Prosthesis Experiences Scales (TAPES). Depressive symptoms were a significant predictor of decreased performance.
39	Rosenbaum (2008, Germany) [62]	Physical activity levels after limb salvage surgery are not related to clinical scores—Objective activity assessment in 22 patients after malignant bone tumor treatment with modular prostheses	Assess PA levels with two objective measurement devices.	*n* = 22(8); 34.5 ± 18.4; 18 TFA, 4 TTA; all tumor	3 DynaPort ADL (2 waist, 1 thigh), Step Activity Monitor (developer N/A); 7 days with SAM and 1st day with DynaPort	Steps per day; steps per weekday and weekend day (for each participant); duration intensity intervals 1–10, 11–20, 21–30, 31–40, 41–50, >50 steps/min (minutes and %); duration lying, sitting, standing, locomotion, undefined (%); movement intensity during walking (m/s^2^); physical activity index	Participants took 4.786 ± 1.770 steps per day. Sitting activity accounted for 54 ± 18% of the recorded time, followed by standing (27 ± 16%), locomotion (10 ± 6%) and lying (8 ± 6%). No correlation between clinical scores and step count measures.
40	Bussmann (2008, The Netherlands) [63]	Daily physical activity and heart rate response in people with a unilateral traumatic transtibial amputation	Investigate if people with unilateral traumatic TTA are less active than people without an amputation, and explore if both groups have a similar heart rate response while walking.	*n* = 9(0); 55.4 (range 21–73); all TTA; all trauma	2 uniaxial, 1 biaxial ADX202 (TEMEC Instruments); 2 upper leg, 2 sternum; 2 days	Duration dynamic activities, walking, dynamic activities besides walking (%); sit-to-stand transitions (*n*); overall and walking body motility (g); resting heart rate; absolute heart rate during walking, normalized heart rate during walking (bpm); heart rate reserve (%)	Participants with amputation had lower percentage dynamic activities and body motility during walking than controls. No significant differences in heart rate and percentage heart rate reserve during walking.
41	Stepien (2007, Australia) [4]	Activity levels among lower-limb amputees: Self-report versus Step Activity Monitor	Determine the accuracy of self-reported activity.	*n* = 77(17); 60 ± 15; 54 TTA, 23 TFA; 39 trauma, 23 vascular, 15 other	StepWatch 3; prosthesis; 8 days	Steps per day; duration rest, low (1–15 steps/min), medium (16–40 steps/min), high (40+ steps/min) intensity activity (%)	Strong agreement between self-reported and measured activity between 9.00am–9:00pm for 34% of participants. Poor agreement between self-reported and measured time spent in various activity intensities.
42	Kanade (2006, UK) [64]	Risk of plantar ulceration in diabetic patients with single-leg amputation	Explore plantar loading of the surviving foot within a wider context of daily walking activity to investigate the precise risk to the surviving limb.	*n* = 21(2); 62.9 ± 6.2; all TTA; all diabetes	StepWatch (Prosthetic Research Study); prosthetic leg; 8 days	Steps per day; daily plantar cumulative stress (DPCS) (MPa/day)	The amputee group walked 30% slower, had reduced cadence, shorter strides and less steps per day than controls without amputation.
43	Kanade (2006, UK) [65]	Walking performance in people with diabetic neuropathy: benefits and threats	Evaluate walking activity on the basis of capacity, performance and potential risk of plantar injury.	*n* = 22(2); 62.9 ± 6.1; all TTA; all diabetes	StepWatch (Prosthetic Research Study); prosthetic leg; 8 days	Steps per day	Total heart beat index increased. Gait velocity and daily stride count fell with progression of foot complications.
44	Hopyan (2006, Australia) [66]	Function and upright time following limb salvage, amputation, and rotationplasty for pediatric sarcoma of bone	Determine the relative physical and psychosocial merits of limb-sparing reconstruction, above-knee amputation, and rotationplasty in survivors of childhood and adolescent lower extremity bone sarcoma.	*n* = 45(23) (20 with amputation); 26 ± 7; 19 TFA, 1 TTA; 20 limb salvage, 19 TFA, 5 rotationplasty, 1 TTA	Uptimer device; thigh; 24-h of weekend day	Uptime (%)	Uptime was highest in persons with rotationplasty, and similar between persons with limb-sparing reconstruction and above-knee amputation.
45	Bussmann (2004, The Netherlands) [67]	Daily physical activity and heart rate response in people with a unilateral transtibial amputation for vascular disease	Study the activity level and heart rate response, objectively measured during normal daily life.	*n* = 9(1); 55 (range 44–76); all TTA; all vascular	2 uniaxial, 1 biaxial ADX202 (TEMEC Instruments); 2 upper leg, 1 sternum; 2 days	Duration dynamic activities, walking, (%); sit-to-stand transitions (*n*); overall and walking body motility (g); resting heart rate; absolute heart rate during walking; normalized heart rate during walking (bpm); percentage heart rate reserve (%)	Participants with amputation had lower activity levels and body motility during walking than controls. No differences in normalized heart rate during walking.
46	Coleman (1999, USA) [21]	Step activity monitor: long-term, continuous recording of ambulatory function	Provide guidelines for use of the Step Activity Monitor (SAM), and results of accuracy and reliability testing, and case study descriptions.	*n* = 2(1); age N/A; 2 TTA; etiology N/A	Step Activity Monitor (later StepWatch); ankle; 2 × 1 week	Total steps; duration inactivity (hours/day), low, moderate and high activity	SAM is accurate, reliable, and can be used to perform long-term step counting on a range of subjects. It is viable means for monitoring gait activity outside of the laboratory during normal daily activities.
**Interventional studies**							
1	Vanicek (2021, UK) [68]	STEPFORWARD study: a randomized controlled feasibility trial of a self-aligning prosthetic ankle-foot for older patients with vascular-related amputations	Determine the feasibility of a Randomized Controlled Trial (RCT) of the effectiveness and cost-effectiveness of a self-aligning prosthetic ankle-foot compared with a standard prosthetic ankle-foot.	*n* = 55(8); 68.8 ± 9.6; all TTA; all non-traumatic (diabetes, PVD, blood clot, or other)	ActivPAL4; prosthesis; 2 × 1 week)	Steps per day; stepping (min/day) (baseline, final)	The consent, retention and completion rates demonstrate that it is feasible to recruit and retain participants to a future trial.
2	Kaluf (2021, USA) [69]	Hydraulic- and microprocessor-controlled ankle-foot prostheses for limited community ambulators with unilateral amputation: pilot study	Examine the benefit of hydraulic- and microprocessor-controlled prosthetic ankles.	*n* = 1(0); 58; TTA; trauma	StepWatch; prosthesis; 3 × 2 weeks	Steps per day; cadence; cadence variability; daily distance, stance/swing time; modus index; ambulation energy index; peak performance index	The four treatments had a varying level of benefits. The hydraulic ankle scored highest in patient-reported outcome measures and step activity data.
3	Kim (2021, USA) [70]	The influence of powered prostheses on user perspectives, metabolics and activity: a randomized crossover trial	Quantify differences between powered and unpowered prostheses and explore relationships between perceptions and functional outcomes in-lab and daily life.	*n* = 10(0); 52.6 ± 11.3; all TTA; 7 trauma, 3 vascular	2 ActiGraph GT9X Link; prosthetic foot and pylon; 2 weeks	Steps per day; steps per day away from home; walking speed (m/s)	No universal benefits of the powered prosthesis. However, effect were subject-specific, and self-reported preferences did not often correlate with objective measures.
4	Sasaki (2020, Thailand) [71]	Sustainable development: a below-knee prostheses liner for resource limited environments (technical briefs)	Develop an affordable ethyl-vinyl-acetate roll-on (AERO) liner for resource-limited environments.	*n* = 1(0); 28; TTA; congenital	Omron HJ-329 Pedometer; prosthetic liner; 2 × 30 days	Steps per day	AERO liner results in increased comfort and speed, and slightly higher residuum temperature. Step count was similar to thermoplastic elastomer (TPE) liner.
5	Miyata (2020, Thailand) [72]	Sustainable, affordable and functional: reimagining prosthetic liners in resource limited environments	Evaluate function and performance of an affordable liner in three types of socket designs.	*n* = 5(2); 60.2 ± 7.4; all TTA; all trauma	Omron HJ-329 Pedometer; in pocket on prosthetic side; 2 × 30 days	Steps per day	AERO liner was suitable for use in both resource limited environments and developed settings standard of care prosthetic treatments.
6	Halsne (2020, USA) [73]	The effect of prosthetic foot stiffness on foot-ankle biomechanics and foot stiffness perception in people with transtibial amputation	Determine the effect of commercial prosthetic foot stiffness category on foot-ankle biomechanics, gait symmetry, community ambulation and relative foot stiffness perception.	*n* = 17(0); 51.0 ± 14.6; all TTA; 11 trauma, 3 dysvascular, 2 infection, 1 other	StepWatch 2; prosthesis; 3 × 2 weeks	Steps per day	Prosthetic foot stiffness category was significantly associated with changes in prosthetic foot-ankle biomechanics, but not with changes in gait symmetry, community ambulation and relative foot stiffness perception.
7	Gaunaurd (2020, USA) [74]	The effectiveness of the DoD/VA mobile device outcomes-based rehabilitation program (MDORP) for high functioning service members and veterans with lower limb amputation	Determine if the MDORP improved strength, mobility and gait quality.	*n* = 17(5); 39.5 ± 11.6; 12 TTA, 4 TFA/KD, 1 bilateral TTA; 14 trauma, 2 infection, 1 cancer	Rehabilitative Lower Limb Orthopedic Analysis Device (ReLOAD) with 5 IMUs; 2 on shank, 2 on thigh and 1 at the sacrum; 8 weeks	Decreased balance, decreased toe load, decreased knee flexion or no deviation (machine learning-derived classifier) (only reported for 1 exemplar participant).	Significant improvements in hip extensor strength, basic and high-level mobility, musculoskeletal endurance, and gait quality after 8–weeks MDORP.
8	Christiansen (2020, USA) [75]	Biobehavioral intervention targeting physical activity behavior change for older veterans after nontraumatic amputation: A randomized controlled trial	Test feasibility of a biobehavioral intervention designed to promote PA.	*n* = 31(0); 65.7 ± 7.6; 26 TTA, 5 TFA; all dysvascular	ActiGraph GT3X-BT; waist; 10 days	Steps per day	The intervention resulted in acceptable participant retention, low dose goal attainment, high participant acceptability, and low safety risk
9	Annis (2019, USA) [76]	Can improved prosthetic alignment increase activity level in patients with lower-extremity amputations? (conference abstract)	Determine if alterations in prosthetic alignment correlate with objective and subjective changes in activity level, function and pain and prosthetic satisfaction.	*n* = 9(1); age N/A; all TTA; 4 trauma, 4 dysvascular, 1 infection	FitBix Flex; prosthesis; 3 weeks	Steps per week	Smart pyramid-guided alignment showed less favorable functional outcomes; recommendations must be used in conjunction with current transtibial dynamic alignment protocols.
10	Littman (2019, USA) [77]	Pilot randomized trial of a telephone-delivered physical activity and weight management intervention for individuals with lower extremity amputation	Test feasibility, acceptability and safety of a weight management and PA intervention and obtain preliminary efficacy estimates for changes in weight, body composition, and physical functioning.	*n* = 15(4); 56.5 ± 11.0; 14 below knee (TTA or toe level), 1 above knee; 6 infection, 5 trauma, 1 cancer, 2 other	StepWatch; prosthesis; 2 × 7 days	Steps per day; sedentary time (hours/day)	Coached participants had greater decreases in waist circumference than the self-directed control group. The home-based intervention was promising in terms of efficacy, safety and acceptability.
11	Morgan (2018, USA) [78]	Laboratory- and community-based health outcomes in people with transtibial amputation using crossover and energy-storing prosthetic feet: A randomized crossover trial	Assess the effects of XF (crossover feet) and ESF (energy storing feet) on health outcomes.	*n* = 27(5); 42.3 ± 11.0; all TTA; 20 trauma, 2 infection, 1 cancer, 4 other	StepWatch; prosthesis; 2 × 4 weeks	Steps per day	XF users experienced improvements in mobility, fatigue, balance confidence, activity restrictions, and functional satisfaction, and exhibited longer sound steps compared to ESF.
12	McDonald (2018, USA) [79]	Energy expenditure in people with transtibial amputation walking with crossover and energy storing prosthetic feet: A randomized within-subject study	Compare energy expenditure at slow, comfortable, and fast walking speeds with XF (crossover feet) and ESF (energy storing feet).	*n* = 27(5); 42.3 ± 11.0; all TTA; 20 trauma, 2 infection, 1 cancer, 4 other	StepWatch; prosthetis; 2 × 4 weeks	Steps per day	Lower oxygen consumption with the XF compared to ESF at each self-selected walking speed, but this was not significant.
13	Kaufman (2018, USA) [80]	Functional assessment and satisfaction of transfemoral amputees with low mobility (FASTK2): A clinical trial of microprocessor-controlled vs. non-microprocessor-controlled knees	Determine if limited community ambulators would benefit from a microprocessor-controlled knee	*n* = 50(22); 69.0 ± 9.0; all TFA; 25 PVD, 13 infection, 5 trauma, 4 thrombosis, 2 cancer, 1 blood disorder	4 ActiGraph GT3X+; waist, thigh and ankles; 4 days	Duration sitting, time up-right activity (%); gait entropy	Improved outcomes with a microprocessor-controlled knee, i.e. fall reduction, less time sitting, and increased activity level. Participants reported significantly improved ambulation, appearance, and utility.
14	Christiansen (2018, USA) [81]	Behavior-change intervention targeting physical function, walking, and disability after dysvascular amputation: A randomized controlled pilot trial	Determine preliminary efficacy of a home-based behavior-change intervention designed to promote exercise, walking activity, and disease self-management.	*n* = 38(3); 63.5; all TTA; all dysvascular	ActiGraph GT3X-BT; waist; 10 days	Steps per day; duration sedentary, light, and moderate/vigorous intensity (%)	The behavior-change intervention group showed within-group increase in daily step count, and had a higher increase in daily step count than the control group, demonstrating that the intervention might increase walking activity.
15	Wurdeman (2017, USA) [82]	Step activity and 6-Minute Walk Test outcomes when wearing low-activity or high-activity prosthetic feet	Determine changes in daily step count and 6MWT with Low-Activity feet (LA) and high-activity Energy-Storage-And-Return (ESAR) feet, and examine sensitivity of these measures to classify different feet.	*n* = 28(N/A); 53.6 ± 11.3; all TTA (4 bilateral); 16 trauma, 8 dysvascular/diabetes, 2 cancer, 2 infection	ActiGraph GT3X-BT; pylon; 2 × 3 weeks	Steps per day	Performance on the 6MWT and daily step counts were similar with the LA and ESAR foot. Correct classification for the 6MWT and step count were 51.9% and 61.5% for the ESAR, and 50% and 50% for the LA foot.
16	Sanders (2017, USA) [83]	Effects of socket size on metrics of socket fit in trans-tibial prosthesis users	Conduct a preliminary effort to identify quantitative metrics to distinguish a good socket from an oversized socket.	*n* = 9(2); 54.1 ± 15.9; all TTA; 6 trauma, 2 congenital, 1 infection	ActiGraph GT3X+; prosthesis; 2 × 4 weeks	Duration activity (hours/day)	Visual analysis showed largest effects for step time asymmetry, step width asymmetry, anterior and anterior-distal morning-to-afternoon fluid volume change, socket comfort scores, and self-reported utility, satisfaction, and residual limb health.
17	Imam (2017, Canada) [84]	A randomized controlled trial to evaluate the feasibility of the Wii Fit for improving walking in older adults with lower limb amputation	Assess the feasibility of Wii.n.Walk for improving walking capacity.	*n* = 28(10); median 62 (range 50–78); 15 TTA, 13 TFA/KD; 15 trauma, 12 dysvascular, 1 cancer	StepWatch, prosthesis; 3 × 1 week	Steps per day	Feasibility of the Wii.n.Walk showed a medium effect size for improving walking capacity.
18	Andrysek (2017, Chile) [85]	Long-term clinical evaluation of the automatic stance-phase lock-controlled prosthetic knee joint in young adults with unilateral above-knee amputation	Evaluate the Automatic Stance-Phase Lock (ASPL) knee mechanism against participants’ existing Weight-Activated Braking (WAB) prosthetic knee.	*n* = 10(4); 20.9 ± 3.1; all TFA; 5 disease, 4 trauma, 1 congenital	Power Walker EX-510; socket; duration N/A	Steps per week	Energy expenditure was lower for ASPL than WAB, but walking speed and step counts were similar. ASPL preference attributed to knee stability and improved walking, while limitations included noise.
19	Klute (2016, USA) [86]	Prosthesis management of residual-limb perspiration with subatmospheric vacuum pressure	Compare a Dynamic Air Exchange (DAE) prosthesis designed to expel accumulated perspiration with a total surface bearing Suction socket that cannot.	*n* = 5(N/A); 44 ± 15; all TTA; 3 trauma, 2 infection	StepWatch; placement N/A; 2 × 1 week	Steps per day	No difference in step activity levels, skin temperatures, and participants’ receptiveness between prostheses.
20	Highsmith (2016, USA) [87]	Effects of the Genium knee system on functional level, stair ambulation, perceptive and economic outcomes in transfemoral amputees	Determine if laboratory determined benefits of Genium are detectable using common clinical assessments and if there are economic benefits with its use.	*n* = 29(4); 46.5 ± 14.2; all TFA; 14 trauma, 4 malignancy, 2 dysvascular	StepWatch; prosthesis; 2 weeks	Galileo-derived K-level	Genium use improved stair walking, multi-directional stepping, functional level, and perceived function. Genium was preferred, and while more costly, the improvements may be worth funding.
21	Raschke (2015, USA) [88]	Biomechanical characteristics, patient preference and activity level with different prosthetic feet: A randomized double blind trial with laboratory and community testing	Measure sagittal moments during walking with three prosthetic feet categories: stiff, intermediate, and compliant forefoot stiffness.	*n* = 11(1); 57 ± 18; all TTA; etiology N/A	StepWatch; placement N/A; 3 × 7 days	Galileo-derived K-level	Participants preferred compliant stiffness. Compliant and intermediate feet had 15% lower maximum sagittal moments, but activity level was not significantly different between feet.
22	Hafner (2015, USA) [89]	Physical performance and self-report outcomes associated with use of passive, adaptive, and active prosthetic knees in persons with unilateral, transfemoral amputation: Randomized crossover trial	Assess and compare physical performance and self-reported outcomes of prosthetic knees with passive, adaptive, and active control.	*n* = 12(0); 58.8 ± 6.1; all TFA; 10 trauma, 2 tumor	StepWatch 3; prosthetic ankle; duration N/A	Steps per day	Adaptive control significantly improved comfortable TUG time and reported physical function compared to passive control. Active control significantly increased comfortable TUG, fast TUG, ramp times and balance confidence compared with passive control.
23	Segal (2014, USA) [90]	Does a torsion adapter improve functional mobility, pain, and fatigue in patients with transtibial amputation?	Explore effects of a torsion adapter on functional mobility and self-perceived pain and fatigue.	*n* = 10(1); 56 ± 12; all TTA; 5 trauma, 4 dysvascular/diabetes, 1 tumor	StepWatch 3; prosthetic pylon; 7 days	Steps per day; steps in low (<15 steps/min), medium (15–40 steps/min), high (>40 steps/min) intensity	Participants wearing a torsion adapter tended to take more low- and medium-intensity steps per day, and experienced less pain than with a rigid adapter.
24	Buis (2014, UK) [91]	Measuring the daily stepping activity of people with transtibial amputation using the ActivPAL™ activity monitor	Compare general activity during 1 week and detailed activity during 24 h period for Patellar Tendon-Bearing (PTB) and Total Surface Bearing (TSB) sockets.	*n* = 48(8); 55.3 ± 13.4; all TTA; 12 PVD, 36 other	ActivPAL; ankle; 6 days	Steps per day; duration walking (%); duration walking per prosthetic socket (%); mean cadence per prosthetic socket (%)	Despite differences in prosthetic socket design, activity levels were similar for both groups.
25	Theeven (2012, The Netherlands) [92]	Influence of advanced prosthetic knee joints on perceived performance and everyday life activity level of low-functional persons with a transfemoral amputation or knee disarticulation	Assess the effects of two types of Microprocessor-controlled Prosthetic Knee (MPK) joints on perceived performance and everyday life activity level.	*n* = 30(8); 59.1 ± 13.0; 24 TTA, 6 KD; 23 trauma, 6 vascular, 1 tumor	ActiGraph GT1M; waist; 3 × 1 week	Up-time (minutes); activity bouts during up-time (*n*); active-time (% of up-time); activity during up-time (*n*, for total group, and subgroups low, intermediate and high)	Participants report benefitting in their performance from using an MPK, but this was not reflected in the daily activity levels.
26	Gailey (2012, USA) [93]	Application of self-report and performance-based outcome measures to determine functional differences between four categories of prosthetic feet	Determine if self-report and performance-based measurements detect functional differences between four categories of prosthetic feet, and if differences exist between with and without PVD.	*n* = 10(1); 55.8 ± 4.1; all TTA; 5 PVD, 5 other	Step Activity Monitor (developer N/A); prosthetic ankle; 5 × 10–14 days	Steps per day; duration activity (hours/day) (for PVD and non-PVD)	Significant differences between PVD and non-PVD groups in Amputee Mobility Predictor (AMPRO) and 6MWT with the Proprio foot. AMPRO was significantly different between baseline and selected feet in PVD group. No differences in self-report measures, PEQ–13, Locomotor Capabilities Index (LCI), 6MWT and SAM.
27	Klute (2011, USA) [94]	Vacuum-assisted socket suspension compared with pin suspension for lower extremity amputees: effect on fit, activity, and limb volume	Investigate effect of a Vacuum-Assisted Socket Suspension system (VASS) as compared with pin suspension.	*n* = 5(N/A); 56 ± 9; all TTA; 4 trauma, 1 dysvascular	StepWatch 3; placement N/A; 2 × 2 weeks	Total steps	Activity levels and residual limb pistoning were significantly lower with VASS. Participants ranked residual limb health higher, were less frustrated and experienced easier ambulation with pin suspension compared to VASS.
28	Agrawal (2010, Germany) [95]	A comparison of gait kinetics between prosthetic feet during functional activities—symmetry in external work (SEW) approach (Ph.D. Thesis)	Determine gait differences among four prosthetic feet, using the Symmetry in External Work (SEW) approach.	*n* = 11(2); 54.8 ± 7.0; all TTA, 6 PVD; 4 trauma, 1 tumor	StepWatch; placement N/A; 4 × 10–14 days	Steps per day; duration low (1–15 steps/min), medium (16–40 steps/min), high (>40 steps/min) activity (hours); duration inactivity and activity (hours)	SEW were significantly different between the K3 foot and other feet during level walking and decline walking. No difference in steps or activity level.
29	Hafner (2007, USA) [96]	Evaluation of function, performance, and preference as transfemoral amputees transition from mechanical to microprocessor control of the prosthetic knee	Evaluate differences in function, performance, and preference between mechanical and microprocessor prosthetic knee (C-leg) control technologies.	*n* = 17(4); 49.1 ± 16.4; all TFA; 10 trauma, 3 malignancy, 2 infection 1 vascular, 1 dysfunction,	StepWatch 2; placement N/A; 4 × 2 weeks	Steps per day; distance walked per day (meters/day)	Result showed improvements in stairs and hills, reduced frequency of stumbling and falling and a preference for the C-leg compared to the mechanical control prosthetic knee.
30	Darter (2007, USA) [97]	The effects of an integrated motor learning based treadmill mobility and aerobic exercise training program in persons with a transfemoral amputation (Ph.D. Thesis)	Investigate the effects of a home-based multiple speed treadmill exercise program.	*n* = 8(3); 41.4 ± 12.1; all TFA; all trauma/tumor	AMP 331 tri-axial activity monitor; prosthesis; 6 weeks	Steps per day; distance walked (meters/day); speed (meters/min/day)	Pre-training distance increased from 1.200 m/day to 1.537 m/day post-training. Steps per day increased from 2.639 pre-training to about 3.488 post-training. Speed changed little over the course of the training.
31	Klute (2006, USA) [98]	Prosthetic intervention effects on activity of lower-extremity amputees	Investigate the effect of prosthetic interventions on the functional mobility.	*n* = 12(N/A); 54 ± 6; all TTA; 7 trauma, 4 dysvascular, 1 infection	StepWatch 3; prosthetic ankle; 2 × 7 days	Steps per day; duration activity (minutes/weekdays, weekend days and all days); figure including number of bouts (dots), bout duration (x-axis) and cadence (y-axis)	Pylon type and knee type had no effect on daily activity level or activity duration.
32	Hsu (2006, Taiwan) [99]	The effects of prosthetic foot design on physiologic measurements, self-selected walking velocity, and physical activity in people with transtibial amputation	Investigate the physiologic differences during multispeed treadmill walking and PA profiles for the Otto Bock C-Walk foot (C-walk), Flex-Foot, and Solid Ankle Cushion Heel (SACH) foot.	*n* = 8(0); 36 ± 15; all TTA; all trauma	Yamax Digiwalker Pedometer; iliac crest of amputation side; 1 month	Steps per day	C-walk had a trend of improved physiologic responses compared with the SACH. Flex-Foot showed significantly lower percentage of age-predicted maximum heart rate and RPE values compared to C-Walk and SACH.
33	Berge (2005, USA) [100]	Efficacy of shock-absorbing versus rigid pylons for impact reduction in transtibial amputees based on laboratory, field, and outcome metrics	Compare Shock-Absorbing Pylons (SAPs) with a conventional rigid pylon, assess effect on gait mechanics, measure transmitted accelerations in situ, and determine functional outcomes using step counts and questionnaires.	*n* = 15(0); 51 ± 9; all TTA; 10 trauma, 4 dysvascular, 1 infection	StepsWatch 2; prosthetic ankle; 2 × 1 week	Steps per week	The only significant difference was for the prosthetic-side knee angle at initial contact, which was higher with the rigid pylon than the SAP while walking a controlled speed, suggesting SAP is as effective as rigid pylon.
34	Coleman (2004, USA) [101]	Quantification of prosthetic outcomes: Elastomeric gel liner with locking pin suspension versus polyethylene foam liner with neoprene sleeve suspension	Compare transtibial socket suspension systems: the Alpha liner with distal locking pin and the Pe-Lite liner with neoprene suspension sleeve.	*n* = 13(3); 49.4 ± 9.6; all TTA; all trauma	StepWatch; prosthetic ankle; 2 × 2 weeks	Steps per day; duration inactivity, low, moderate, high (>30 steps/min) intensity activity (hours/day); distribution low, moderate, high intensity of active time (%); socket wear time (hours/day)	Participants spent 82% more time wearing the Pe-Lite and took 83% more steps per day. Ambulatory intensity distribution and questionnaire results were not different between sockets.
**Algorithm/method development studies**							
1	Srisuwan (2021, USA) [102]	Locomotor activities of individuals with lower-limb amputation	Describe a novel method for activity monitoring and use it to identify step count distribution of locomotor activities in the home, work, and community environments.	*n* = 10(0); 48.7 ± 17.0; all TTA; 8 trauma, 2 diabetes	Custom instrument including ADXL345 triaxial accelerometer and L3G4200D triaxial rate gyroscope; prosthetic pylon; 34.7 ± 13 h	Total number of steps; steps per activity classifications straight, turn right/left, stair up/down, ramp up/down, turn prosthetic leg/nonamputated leg inside	The method can be used to accurately classify locomotor activities in home, work, and community environments.
2	Jamieson (2021, UK) [103]	Human activity recognition of individuals with lower limb amputation in free-living conditions: A pilot study	Investigate the implementation of supervised classifiers and a neural network for the recognition of activities.	*n* = 4(1); 50 ± 7.7; 3 TTA, 1 bilateral TTA; 3 trauma, 1 vascular	ActivPAL4+; chest-mounted camera, GPS on iPhone 6; thigh; 7 days	5-Fold accuracy per model (%); 5-Fold accuracy per level label (%); F1 scores per level label resolution; Leave One Subject Out (LOSO) accuracy (%, for each participant); confusion matrices (per level label resolution and model)	The models Support Vector Machine (SVM) and Long Short-Term Memory Network (LSTM) showed 77–78% classification accuracy, but fell with increased label detail. Classifiers trained on individuals without gait impairment could not recognize activities carried out by LLAs.
3	Griffiths (2021, UK) [104]	A machine learning classification model for monitoring the daily physical behavior of lower-limb amputees	Develop a model capable of accurately classifying lower-limb amputee postures by using data from a single shank-worn accelerometer.	*n* = 1(N/A); age N/A; bilateral TFA; etiology N/A	2 ActivPAL PAL3; thigh and shank; 7 days	F-scores and confusion matrices for 8 models and 4 posture classes (sitting, standing, stepping, lying)	A random forest classifier with 15–s window length provided a 93% weighted average F-score accuracy, and between 88 and 98% classification accuracy across four posture classes.
4	Kim (2020, USA) [105]	Wearable sensors quantify mobility in people with lower limb amputation during daily life	Explore the clinical viability of using wearable sensors to characterize functional mobility.	*n* = 17(2); 47.9 ± 14.5; 16 TTA, 1 bilateral TTA & TFA; 10 trauma, 6 dysvascular, 1 congenital	2 ActiGraph GR9X Link and GPS enabled smartphone; ankle and foot on prosthesis; 2 weeks	Steps per day; steps per bout; stride length (meters), home and away from home; total IMU straight-line walking strides (with successful GPS match); cadence (strides/min); walking speed (m/s)	Functional capacity measured in the lab was not reflected in routine walking during daily life. This approach can be used to aid in prosthetic prescription or in the assessment of interventions.
5	Weathersby (2018, USA) [106]	Development of a magnetic composite material for measurement of residual limb displacements in prosthetic sockets	Design and evaluate a novel wearable inductive sensor system for long-term measurements of limb-socket displacements.	*n* = 2(0); age N/A; all TTA; all trauma	Custom sensor design including chip, antenna, capacitor and sheath with embedded magnetic particles; liner; 2 and 4 weeks	Signal loss per test location (%)	Field testing demonstrated less than 3% signal degradation after four weeks; the developed sensor meets durability and performance needs and is ready for large-scale clinical testing.
6	Swanson (2018, USA) [107]	Instrumented socket inserts for sensing interaction at the limb-socket interface	Investigate a strategy for designing and fabricating computer-manufactured socket inserts with sensors that measure limb-socket interactions.	*n* = 1(0); 74; TTA; trauma	Custom sensor design including proximity, force sensing resistor and inductive sensor; socket; 2 days	Sensor change (proximity (a.u. × 10^4^, distance (mm); pressure (kPa)); change in sensor behavior (yes, no, minimal)	Multiple sensor types were necessary in analysis of field collected data to interpret how sock changes affected limb-socket interactions.
7	Arch (2017, USA) [108]	Method to quantify cadence variability of individuals with lower-limb amputation	Develop and demonstrate feasibility of a method to quantify real-world cadence variability.	*n* = 27(6); 56.8 ± 12.2; 20 TTA, 7 TFA; etiology N/A	Fitbit One; prosthesis; 7 days	Average cadence; maximum cadence (steps/min); cadence variability scale parameter; cadence variability (Weibul probability density)	K2 walked with significantly less cadence variability than K3. The method was able to differentiate cadence characteristics between K2 and K3 ambulators.
8	Gardner (2016, USA) [109]	Monitoring prosthesis user activity and doffing using an activity monitor and proximity sensors	Develop a method to incorporate doffing and donning information into activity characterization.	*n* = 23(N/A); 55.6 ± 14.9; all TTA; etiology N/A	2 Actigraph GT3X+ (prosthesis, thigh); 2 proximity sensors (socket brim); 3 h	Doffing, sitting, standing and walking (*n*)	Detected activities matched participants’ descriptions of activities well, of which 95% of doffs were detected, making the developed technology relevant to use.
9	Jayaraman (2014, USA) [110]	Global position sensing and step activity as outcome measures of community mobility and social interaction for an individual with a transfemoral amputation due to dysvascular disease	Examine the combined use of GPS and a step activity monitor to quantify community mobility and social interaction.	*n* = 1(1); 76; TFA; dysvascular	StepWatch 3.1 and GPS (Travel Recorder XT); placement N/A; 1 month	Steps per day; steps per day at home; steps per day outside of home; duration at community locations (commercial, religious, other residential, open space, mixed use), and at home (hours and minutes); trips (*n*, car, wheelchair, walking, and undetermined trips)	GPS and step activity monitor provided quantitative details on the patient’s steps taken in and out of the home, wheelchair use, prosthesis use, driving trips, and time spent on social and community trips.
10	Redfield (2013, USA) [111]	Classifying prosthetic use via accelerometry in persons with trans-tibial amputations	Classify the movements and body postures by using commercially-available accelerometers and a custom software algorithm.	*n* = 8(2); 53.0 ± 11.6; all TTA; 8 trauma, 1 infection, 1 tumor	2 ActiGraph GT3X+; prosthesis and thigh; 2 days	Agreement between activity classifications (doffed, sitting, standing, active) (%)	The classifier achieved a mean accuracy of 96.6%.
11	Frossard (2011, Australia) [112]	Categorization of activities of daily living of lower limb amputees during short-term use of a portable kinetic recording system: A preliminary study	Determine the relevance of the categorization of the load regime data to assess the functional output and prosthetic use.	*n* = 1(0); 33; TFA; etiology N/A	Custom instrument, including multiaxial transducer; prosthesis; 5 h	Number of steps; cadence; activity categories directional locomotion, localized locomotion, stationary loading, inactivity (*n*, hours, %); duration gait cycle, swing and support phases (seconds); forces (N), moments (Nm), impulse (kN·s) along anteroposterior, mediolateral and long axis (per activity category)	The proposed categorization and apparatus have the potential to complement conventional instrument, particularly for difficult cases.
12	Frossard (2008, Australia) [113]	Monitoring of the load regime applied on the osseointegrated fixation of a trans-femoral amputee: A tool for evidence-based practice	Describe the continuous recording of the true load regime experienced during daily living by the abutment of a transfemoral amputee fitted with an osseointegrated fixation.	*n* = 1(0); 33; TFA; etiology N/A	Set Activity Monitor Pedometer, and custom instrument, including multiaxial transducer; prosthesis; 5 h	Number of steps; cadence; duration gait cycle, swing and support phases (seconds); activity and inactivity (%); forces (N) and moments (Nm) on the anteroposterior, mediolateral and long axes of the abutment	The overall load profile presented variable length of inactivity (64%) and activity (36%). The maximum load applied on the mediolateral, anteroposterior, and the long axes represented 21%, 21% and 120% of the body weight, respectively.
**Validity/Feasibility studies**							
1	Godfrey (2018, USA) [114]	The accuracy and validity of Modus Trex Activity Monitor in determining functional level in veterans with transtibial amputations	Investigate the accuracy and reliability of Modus Trex-derived K-level to differentiate between Medicare Functional Classification levels (K-levels).	*n* = 27(0); 60.5 ± 18.6; all TTA; 13 trauma, 11 dysvascular, 3 other	StepWatch and GPS device; prosthesis; 2 weeks	Steps per day; peak cadence; Modus Trex-derived K-level and Modified Clinical K-level	The Modus Trex-derived K-level correlated most strongly with the MCK-levels.
2	Arch (2018, USA) [115]	Step count accuracy of StepWatch and FitBit One among individuals with a unilateral transtibial amputation	Evaluate the step count accuracy of both monitors during forward-linear and complex walking and compare monitor step counts in the free-living environment.	*n* = 50(N/A); age N/A; all TTA; etiology N/A	StepWatch and Fitbit One; prosthesis; 7 days	Total steps (for each participant)	Both monitors accurately counted steps during forward linear walking, StepWatch was more accurate than FitBit during complex walking, and FitBit counted fewer steps than StepWatch during free-living walking.
3	Orendurff (2016, USA) [116]	Comparison of a computerized algorithm and prosthetists’ judgment in rating functional levels based on daily step activity in transtibial amputees	Compare prosthetists’ ratings of functional levels based on a visual inspection of step activity patterns with the ratings calculated by the computerized algorithm based on the same step activity.	*n* = 14(N/A); age N/A; all TTA; etiology N/A	StepWatch; placement N/A; 5–7 days	Calculated K-level; prosthetists’ rating of K-level	The computerized algorithm produced functional level closely matched the average of the rating by the 14 experienced prosthetists.
4	Albert (2014, USA) [117]	Monitoring daily function in persons with transfemoral amputations using a commercial activity monitor: A feasibility study	Assess mobility using data collected from a popular, consumer-oriented activity monitor Fitbit.	*n* = 9(4); 53 ± 12; all TFA; 6 trauma, 2 vascular, 1 cancer	Fitbit One; waist; 7 days	Steps per day; duration lightly active, fairly active, very active (%); duration total daily activity (%); calories; Fitbit activity score; miles walked; floors climbed (*n*)	Fitbit results correlate with K-level, and Fitbit activity score is independent of variations in age, weight, and height compared with estimated calories.
5	Albert (2013, USA) [118]	Monitoring functional capability of individuals with lower limb amputations using mobile phones	Provide evidence that accelerometry using mobile phones can be used to objectively quantify activity levels.	*n* = 10(5); 53.1 ± 11.9; all TFA; 7 trauma, 2 dysvascular, 1 cancer	Mobile phone; in belt in center of the back; 7 days	Duration low (0.1–0.5 m/s^2^), medium (0.5–1.0 m/s^2^), high (1.0+ m/s^2^) activity	K-level was correlated to the proportion of moderate to high activity, which suggests that mobile phones can be used to evaluate real world activity for mobility assessment.
6	Van Dam (2001, The Netherlands) [119]	Measuring physical activity in patients after surgery for a malignant tumour in the leg. The reliability and validity of a continuous ambulatory activity monitor	Investigate reliability and validity of an ambulatory activity monitor in measuring the intensity of PA in patients who underwent radical surgery for a malignant tumour in the femur or tibia.	*n* = 20(10) (8 with amputation); 49 (median, range 18–69); 3 TFA, 3 KD, 2 rotationplasty; all tumor	3 Dynaport ADL; 2 waist and 1 thigh; 2 × at least 24 h.	Duration walking, standing, sitting (%); movement intensity of walking, standing, sitting (m/s^2^); test-rest reliability (ICC values)	Reliability was satisfactory, with intraclass correlation coefficients ranging from 0.65 to 0.91.

Physical Activity (PA), Trans-Tibial Amputation (TTA), Trans-Femoral Amputation (TFA), Knee Disarticulation (KD), Lower Limb Amputation (LLA), Not Applicable (N/A), Peripheral Vascular Disease (PVD), Amputee Mobility Predictor (AMP), Timed Up and Go (TUG), 2 Minute Walk Test (2MWT), Step Activity Monitor (SAM), Prosthesis Evaluation Questionnaire (PEQ), Global Positioning System (GPS), Preferred walking speed (PWS), 6 Minute Walk Test (6MWT).

**Table 2 sensors-22-02243-t002:** Number (percentage) of reported outcome measures categories per study design and total.

Study Design	No. of Studies	Step Count	Fitness/ Intensity Activity	Type Activity/ Posture	Commercial Scores	Prosthetic Use/Fit	Gait Quality	GPS	Accuracy
Observational	46	36 (78)	15 (33)	12 (26)	4 (9)	6 (13)	1 (2)	2 (4)	0
Interventional	34	28 (82)	8 (24)	9 (26)	7 (21)	2 (6)	3 (9)	1 (3)	0
Algorithm/method development	12	5 (42)	3 (25)	4 (33)	1 (8)	3 (25)	3 (25)	1 (8)	3 (25)
Validity/Feasibility	6	3 (50)	4 (67)	1 (17)	3 (50)	0	0	1 (17)	1 (17)
Total	98	72 (73)	30 (31)	26 (27)	15 (15)	11 (11)	7 (7)	5 (5)	4 (4)

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
