# Peer review of "Reported Outcome Measures in Studies of Real-World Ambulation in People with a Lower Limb Amputation: A Scoping Review"

_sensors, 2022, doi:10.3390/s22062243_

Round 1

Reviewer 1 Report

Thank you for the review invitation. This scoping review aims to survey the status of wearable sensors usage in monitoring the ambulation function of people with lower-limb amputation (LLA) in the real-world environment. Results of the review show that the number of studies using wearable sensors is increasing. A number of them adopted the step count as the outcome from the sensors. Only a few studies use wearable sensors to assess the gait quality in people with LLA. The authors also conclude that the current technology/studies cannot provide a comprehensive picture of the ambulation function of people with LLA in the real-world environment.

This review seems to be well conducted and provides an overall picture of the utilization of wearable sensors in people with LLA in the real world situation. I find that the interpretation and discussion of results need to be improved and the suggestion of future direction should be more specific and concrete.

My major concern is that the future research direction suggested by the authors is too vague and non-specific. The authors suggested that ‘ecological validity should get more attention in the field of prosthetic mobility.’ However, the authors have not indicated which type of study (study design/methodology) could establish or enhance our understanding of the ‘ecological validity of the wearable sensor. Alternatively, the authors may make use of 1 or 2 reviewed studies as an example to illustrate the preferred approach. I hope that the authors can propose a more concrete research direction based on the results of this review.

Besides, I am confused about the authors’ comments regarding the ecological validity of included studies. For example, the authors suggested that ‘Measurements in the real world may overcome some of the limitations of in-laboratory testing, but the degree of ecological validity of these studies can still be disputed (Line 504).’ I hope that the author can further elaborate on how the ecological validity would be disrupted, given that these studies were already conducted in the real world situations.

When discussing the ecological validity of the studies using wearable sensors in people with LLA, the authors commented that ‘the contemporary technology is limited in providing a comprehensive picture of real-world ambulation, limiting the level of ecological validity.’(Line 27, abstract). However, the authors also stated that ‘monitoring prosthetic ambulation in the real world is in principle high in ecological validity (Line 516). The two standpoints seem contradictory to me.

The authors should also consider using a more generic term to replace the phrase ‘ecological validity’. Several studies have argued that the definition, “the extent to which experimental findings can generalize to the real world situation that a researcher wishes to understand”, is not a correct interpretation of ecological validity. (Please see Kihlstrom 2021; Holleman et al. 2021).

Holleman, G. A., Hooge, I. T. C., Kemner, C., Hessels, R. S. (2021). The reality of “real-life” neuroscience: A commentary on Shamay-Tsoory and Mendelsohn (2019). Perspectives on Psychological Science, 16(2), 461–465

Kihlstrom, J. F. (2021). Ecological validity and “ecological validity”. Perspectives on Psychological Science16(2), 466-471.

Reviewer 2 Report

The topic is promising, however, is recommended to improve the following aspects:

  1. The abbreviations should be declared the first time is used capitalizing the first letter of each word that appears in the abbreviation, for instance: lower limb amputation (LLA)= Lowe Limb Amputation, Chronic obstructive pulmonary disease (COPD)= Chronic Obstructive Pulmonary Disease (COPD), Joanna Briggs Institute (JBI).
  2. 3.3. Review findings 3.3.2. Categories of reported outcome measures. Where is the 3.3.1.? I did not find it.
  3. Line 576: validation, X.X., Y.Y. and Z.Z.; What is this?
  4. Explain why only were used three databases? Missing information of patents?
  5. To clarify the contribution of the manuscript in relation to the Sensor journal scope. To include information about metrics, advantages, error, performance of wearable technology and detail it in the conclusion section.
  6. The conclusion section does not describe with metrics the important results of review nether gives conclusive suggestions of why perhaps are obtained these metrics and the useful in real-world ambulation in people with lower limb amputation.
  7. To detail applications of these findings and future works.

Round 2

Reviewer 1 Report

The authors have addressed all my comments and I am satisfied with the revised manuscript.

Reviewer 2 Report

The authors heeded the recommendations and now the manuscript shows the contribution in a better way.